# PV-Tuning: Beyond Straight-Through Estimation for Extreme LLM Compression

**Vladimir Malinovskii**[†]
Yandex, HSE University

**Denis Mazur**[†]
MIPT[◇], SberDevices[¶]

**Ivan Ilin**[†]
AI Initiative, KAUST[*]

**Denis Kuznedelev**
Yandex, Skoltech

**Konstantin Burlachenko**
AI Initiative, KAUST[*]

**Kai Yi**
AI Initiative, KAUST[*]

**Dan Alistarh**[‡]
IST Austria, NeuralMagic

**Peter Richtarik**[‡]
AI Initiative, KAUST[*]

## Abstract

There has been significant interest in "extreme" compression of large language models (LLMs), i.e., to 1-2 bits per parameter, which allows such models to be executed efficiently on resource-constrained devices. Existing work focused on improved one-shot quantization techniques and weight representations; yet, purely post-training approaches are reaching diminishing returns in terms of the accuracy-vs-bit-width trade-off. State-of-the-art quantization methods such as QuIP# and AQLM include fine-tuning (part of) the compressed parameters over a limited amount of calibration data; however, such fine-tuning techniques over compressed weights often make exclusive use of *straight-through estimators (STE)*, whose performance is not well-understood in this setting. In this work, we question the use of STE for extreme LLM compression, showing that it can be sub-optimal, and perform a systematic study of quantization-aware fine-tuning strategies for LLMs. We propose PV-Tuning — a representation-agnostic framework that generalizes and improves upon existing fine-tuning strategies, and provides convergence guarantees in restricted cases. On the practical side, when used for 1-2 bit vector quantization, PV-Tuning outperforms prior techniques for highly-performant models such as Llama and Mistral. Using PV-Tuning, we achieve the first Pareto-optimal quantization for Llama-2 family models at 2 bits per parameter.

## 1 Introduction

Recent years have seen the development of ever more capable large language models, attracting immense interest from both researchers and industry. One of the driving factors behind progress in this area is the availability of powerful **open** LLMs such as Llama [69], Mistral [34, 35], or Phi [41]. The main advantage of open LLMs is that they can be run and fine-tuned locally by end users; however, as state-of-the-art LLMs grow larger, they also become harder to run on commodity hardware. For instance, in order to fit the best available Llama-3 model on a consumer GPU, the model would have to be compressed to below 2 bits per parameter[1].

To achieve such "extreme" degrees of compression accurately, researchers have proposed a variety of techniques, which can be roughly categorized into i) better quantized weight representations and ii) better algorithms to learn these representations. The weight representations used for extreme quantization include group quantization [22, 20], sparse high-precision outliers [17, 32], incoherence processing of the weights [9, 70], or additive and residual quantization [21, 72]. In turn, the calibration algorithms also vary between data-free methods [20], layer-wise calibration [22, 18], block-wise or global fine-tuning [21, 71] or even quantization-aware training [78, 75]. However, the

---

[†]Equal contribution.    ‡ Equal senior authors.    ◇ Moscow Institute of Physics and Technology, Russia

[¶] Work performed while at Yandex.    * King Abdullah University of Science and Technology, Saudi Arabia

[1]At the time of writing, the best open model (Llama-3 70B) occupies 130GB in FP16, while most consumer GPUs have 8-24GiB DRAM, some of which must be reserved for the attention cache.

38th Conference on Neural Information Processing Systems (NeurIPS 2024).

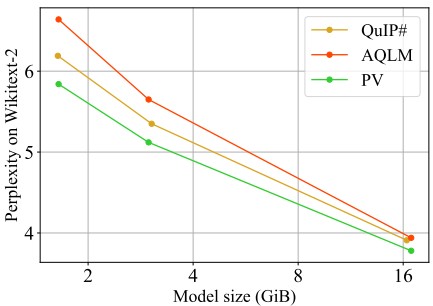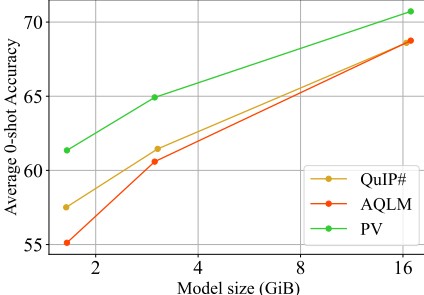

Figure 1: WikiText-2 perplexity (left) and average zero-shot accuracy (right) of 2-bit quantized LLAMA 2 models as a function of model size (GiB). See detailed setup in Section 4.3.

weight representation and the fine-tuning algorithm are largely orthogonal: most popular quantized representations could be obtained layer-wise in one-shot, fine-tuned layer-wise to a variety of optimization objectives, or even trained entirely from scratch.

Surprisingly, there is a clear disparity between the degree of interest shown to accurate one-shot[2] quantization versus accurate fine-tuning. Specifically, one-shot quantization is very well-studied, to the extent that, as shown in Figure 2, improvements in this direction are clearly saturating. At the same time, the impact of fine-tuning strategy is largely unknown: while many recent works use some form of fine-tuning [63, 21, 71], they typically consider a single fine-tuning regimen based on straight-through estimation (STE) [6, 15]. Thus, given the multitude of representations considered, it is not at all clear whether current fine-tuning strategies are optimal.

In this work, we analyze the problem of fine-tuning over highly-compressed weights from the optimization perspective. We begin by analyzing popular fine-tuning strategies for extreme LLM quantization. The key challenge in this context is that the quantized representations may contain both continuous and discrete variables: while continuous parameters, such as learnable scales or codebooks, can be optimized by backpropagation, the discrete parameters (e.g., integer assignments for the weights) cannot. Existing fine-tuning techniques either do not optimize over discrete parameters at all [71, 21] or fine-tune them using heuristics such as STE or stochastic rounding [3]. Unfortunately, these methods are not well-justified for weight quantization from the point of view of optimization theory, and, as we show in Section 3, can provide poor practical performance.

We propose an alternative solution: instead of following heuristic gradient estimates, our approach follows the actual gradient of the objective in a small subspace of optimized parameters where it can be meaningfully improved. Following this insight, we formulate the *PV-tuning framework* for fine-tuning arbitrary quantized representations. We update both discrete and continuous components to minimize a global objective function, such as the KL divergence relative to the original model predictions. Our results show that this strategy leads to significant improvements across weight representations, achieving new state-of-the-art in compression-accuracy trade-offs.

The main contributions of our work can be summarized as follows:

1. We analyze the problem for training discrete quantized representations for better understanding of the limitations of existing optimization algorithms. We then propose a novel algorithm inspired by compressed gradient methods that addresses these limitations. When compared to straight-through estimation and stochastic rounding, our approach 1) can be shown to converge to a stable solution; and 2) this solution is significantly more accurate in practice.

2. We generalize the proposed algorithm into the PV-Tuning framework[3], which can minimize a global objective function over a general quantized representation, by optimizing both continuous and discrete parameters via a variant of coordinate descent.

3. We demonstrate that PV-tuning can improve quantized model accuracy for leading existing approaches, including GPTQ and AQLM, on popular LLMs including Llama-2 & 3 and Mistral. Our procedure achieves state-of-the-art accuracy (measured through perplexity) in 1- and 2-bit quantization regimes while using the same amount of calibration data as the original algorithms. Importantly, the PV-tuned models use the same underlying weight representations, and are compatible with existing inference kernels. In terms of accuracy per model size, PV-tuning of vector quantization outperforms all prior techniques in the 1-3 bits/parameter range, and is the first to achieve Pareto-optimal quantization for Llama-2 models at around 2 bits per parameter.

---

[2]By "one-shot" we refer to methods that quantize quantize the model in a single pass over calibration data.

[3]The official implementation is available at `https://github.com/Vahe1994/AQLM/tree/pv-tuning`.

## 2 Background

**Post-Training LLM Quantization (PTQ).** There has been significant interest in PTQ methods [49, 25] that would scale to LLMs. Early work [17, 78, 50] used direct round-to-nearest (RTN) quantization over weight groups of well-chosen size. GPTQ [22] improved upon these results significantly via an accurate one-shot solver for minimizing layer-wise compression errors. Next, AWQ [42] improved upon these results by employing per-channel scaling to reduce the error on important weights while SqueezeLLM [36] implemented non-uniform quantization. QuIP [9] proposed a more accurate weight representation by leveraging incoherence matrices. Another line of works [18, 39] proposes an improved quantized weight representation, which saves a small fraction of outliers in full precision. Other recent works propose augmenting quantized representations with lowrank "adapters" that compensate quantization error [28, 84]. Recently, BiLLM [32] developed residual binarization that stores salient weights in progressively higher bitwidth, quantizing models to nearly 1 bit per parameter at non-catastrophic accuracy loss.

Currently, the state-of-the-art methods in terms of accuracy-vs-size are QuIP# [71] and AQLM [21]. Both methods work roughly by mapping weight groups to points on highly-dimensional lattices, which are either chosen to satisfy some optimality properties (for QuIP#) or are learned (for AQLM). Interestingly, AQLM showed that fine-tuning the continuous parameters (codebooks) can improve accuracy significantly relative to pure one-shot compression; a variant of this approach was also adopted by QuIP#. PV-Tuning is compatible with both methods: as we show, it can lead to state-of-the-art compression results for such representations.

**Fine-tuning over Quantized Weights.** As mentioned above, the two SOTA quantization techniques apply fine-tuning, but only update *continuous* parameters, such as quantization scales. When optimizing over *discrete* parameter sets, a standard choice in deep learning is the Straight-Through Estimator (STE) [6, 15, 73]. Prior work on LLM compression proposed to update both continuous and discrete parameters, via STE, both for post-training quantization [78, 63] and for training quantized networks from scratch [32]. However, it was observed early on that STE leads to instability when fine-tuning heavily quantized LLMs [78]. While early results suggest that STE can perform well when training quantized models from scratch [44], this behavior is yet to be validated for highly-performant multi-billion-parameter models, which are the focus of our work.

In summary, the two standard approaches for fine-tuning quantized LLMs are 1) fine-tuning only over the continuous parameters, such as quantization scales, which heavily limits the number of trainable parameters; and 2) optimizing all parameters via the STE, which however is known to be quite noisy especially for extreme quantization. In this context, our work proposes alternative approaches in the post-training compression setting, which lead to state-of-the-art results relative to both options.

## 3 Fine-Tuning Quantized Models

In this section, we study the problem of fine-tuning quantized models to minimize a global objective, such as cross-entropy. Section 3.1 formulates this problem from an optimization perspective and introduces our notation. In Section 3.2, we analyze several popular strategies for solving this problem and highlight some of their limitations. To circumvent these limitations, we propose an alternative optimization algorithm in Section 3.3 and discuss implementation details in Section 3.4.

### 3.1 Problem description

Consider the problem of minimizing objective (loss) $\phi$,

$$\min_{x \in \mathbb{R}_c^d} \phi(x), \tag{1}$$

where $\phi : \mathbb{R}^d \to \mathbb{R}$ is a differentiable function bounded from below (e.g., by zero), and $\mathbb{R}_c^d \subset \mathbb{R}^d$ is a set of all possible quantized weights that can be represented with a given quantization method. Without loss of generality[4], we first analyze the case of scalar nonlinear quantization. In this scenario, $c \in [d] := \{1, 2, \ldots, d\}$ (typically $c \ll d$), and $\mathbb{R}_c^d \subset \mathbb{R}^d$ is the set of all vectors in $\mathbb{R}^d$ whose $d$ entries take exactly $c$ distinct values. In other words, the cardinality of the set $V(x) := \{x_1, \ldots, x_d\}$ is equal to $c$, and we can therefore write $\mathbb{R}_c^d := \{x \in \mathbb{R}^d : |V(x)| = c\}$.

---

[4]We explain how this generalizes to other quantized representations in Appendix C

**Useful notation.** A vector $x \in \mathbb{R}_c^d$ naturally induces a partition, which we shall call $P(x)$, of the set $\{1, \ldots, d\}$ into $c$ nonempty subsets $P_1(x), \ldots, P_c(x)$ characterized by

$$x_i = x_j \quad \Leftrightarrow \quad \exists k \ : \ i \in P_k \text{ and } j \in P_k.$$

Let's denote $P(x) := \{P_1(x), \ldots, P_c(x)\}$. Moreover, we shall write $P(y) \sqsupseteq P(x)$ if each element of $P(x)$ is a subset of some element of $P(y)$. For distinct $i, j \in [d]$, let us introduce the notation $\delta_{ij}(x) = 1$ if there exists $k$ such that $i, j \in P_k(x)$, and $\delta_{ij}(x) = 0$ otherwise. Given this notation, notice that $P(y) \sqsupseteq P(x)$ if and only if for all $i \neq j$ we have $\delta_{ij}(x) = 1 \Rightarrow y_i = y_j$. Finally, we define $\mathbb{R}_{\leq c}^d := \mathbb{R}_1^d \cup \cdots \cup \mathbb{R}_c^d$ as the set of all vectors in $\mathbb{R}^d$ whose $d$ entries take at most $c$ distinct values. So, if $x \in \mathbb{R}_c^d$ and $P(y) \sqsupseteq P(x)$, then $y \in \mathbb{R}_{\leq c}^d$.

**PV method.** Following this notation, we define an optimization algorithm that alternates between optimizing $\phi$ with fixed $P$ or fixed $V$. From a practitioner's point of view, these represent optimizing continuous parameters (scales, codebooks, zeros) and discrete codes (assignments), respectively.

◇ **The P step (fixing $P$).** Given $x \in \mathbb{R}_c^d$, consider the mapping

$$M_P(x) = M_{P,\phi}(x) := \arg \min_{y \in \mathbb{R}^d} \{\phi(y) \ : \ P(y) \sqsupseteq P(x)\}. \tag{2}$$

Notice that, necessarily, $M_P(x) \in \mathbb{R}_{\leq c}^d$ and $\phi(M_P(x)) \leq \phi(M_P(x)) \leq \phi(x)$. Evaluating $M_P$ amounts to solving an unconstrained optimization problem in a $c$-dimensional space.

◇ **The V step (fixing $V$).** Similarly, given $y \in \mathbb{R}_c^d$, we define the mapping

$$M_V(y) = M_{V,\phi}(y) := \arg \min_{x \in \mathbb{R}^d} \{\phi(x) \ : \ V(x) \subseteq V(y)\}. \tag{3}$$

Likewise, $M_V(y) \in \mathbb{R}_{\leq c}^d$ and $\phi(M_V(y)) \leq \phi(M_V(y)) \leq \phi(y)$. Evaluating $M_V$ amounts to solving difficult discrete optimization problems with a search space of size $|V(x)|^d \leq c^d$ (exponential in $d$).

---

**Algorithm 1** PV algorithm

---

1: **Initialization:** starting point $x^0 \in \mathbb{R}_{\leq c}^d$
2: **for** $k = 0, 1, \ldots$ **do**
3: $\quad y^k = M_P(x^k) := \arg\min_{y \in \mathbb{R}^d} \left\{\phi(y) : P(y) \sqsupseteq P(x^k)\right\}$ $\qquad\qquad$ (P step: continuous)
4: $\quad x^{k+1} = M_V(y^k) := \arg\min_{x \in \mathbb{R}^d} \left\{\phi(x) : V(x) \subseteq V(y^k)\right\}$ $\qquad\qquad$ (V step: discrete)
5: **end for**

---

Our key algorithmic idea, in its simplest form, is to optimize $\phi$ by alternating the P and V steps, i.e., iteratively applying the $M_P$ and $M_V$ operators. (We will propose several more practically-useful approximations and variations later; see Sections 3.2–3.3 and also Appendix B.) This resulting method, which we call the PV method, is formalized as Algorithm 1. Our key guarantee for the PV method is formalized in the next result.

**Theorem 3.1** (Convergence of the PV method). *Assume $\phi$ is bounded below, and let $x^0 \in \mathbb{R}_c^d$. Then (i) $y^k \in \mathbb{R}_{\leq c}^d$ and $x^k \in \mathbb{R}_{\leq c}^d$ for all $k \geq 0$; (ii) $\phi(x^{k+1}) \leq \phi(y^k) \leq \phi(x^k)$ for all $k \geq 0$; and (iii) the sequence $\{\phi(x^k)\}_{k \geq 0}$ converges.*

The proof can be found in Appendix A.1. Note that we do not claim that the method converges to a minimizer of $\phi$; the optimization problem is too difficult for us to be able to guarantee this. However, as we shall see in the numerical results, we nevertheless obtain great empirical performance, especially when coupling the PV approach with some additional algorithmic tricks.

This general approach is popular in "shallow" machine learning problems; for instance, if $\phi(x) = \|x - z\|^2$ is the squared error with respect to some user-specified vector $z$, then the above algorithm recovers 1-dimensional $K$-means on the data vector $z$. Likewise, if $\phi(\cdot)$ is the log-likelihood, then, depending on the choice of the set $\mathbb{R}_c^d$, the approach is related to the EM algorithm [16].

In turn, we apply the PV method to obtaining highly-accurate quantized LLMs. Applying the PV method "as is", would be infeasible in practice: computing the P and V mappings requires solving difficult optimization problems especially due to LLM parameter scales. However, both mappings can be approximated. The P step can be reparameterized as an unconstrained optimization problem on the

unique values in the weight matrix. Practically it means that the "codebooks" can be optimized using an automated differentiation engine (i.e. PyTorch). However, for many quantized representations, $M_P(x)$ can be approximated by one or more steps of GD, directly optimizing $\phi$ over the set $V(x)$ of its $c$ unique values. The $c$-dimensional gradient can be computed efficiently by backprop, as described in prior works [63, 71]. On the other hand, the V step ($M_V(\cdot)$) is more difficult to approximate as it involves searching a discrete space of size $c^d$. We dedicate the next two sections to this task.

## 3.2 Linearized V step & gradient-based discrete updates

The V mapping (3) can be approximated by solving a discrete least squares problem using an approximation of $\phi(x)$ around $y$:

$$\phi(x) \approx \widetilde{\phi}_y(x) := \phi(y) + \langle \nabla \phi(y), x - y \rangle + \frac{L}{2} \|x - y\|^2, \tag{4}$$

where $L > 0$ is a sufficiently large constant. Subsequently, we perform the V step using the simpler convex quadratic function $\widetilde{\phi}_y$ instead of the typically more complicated function $\phi$:

$$M_{V,\phi}(y) \overset{(4)}{\approx} M_{V,\widetilde{\phi}_y}(y) \overset{(3)}{=} \arg\min_{x \in \mathbb{R}^d} \left\{ \widetilde{\phi}_y(x) \; : \; V(x) \subseteq V(y) \right\}.$$

Our first lemma shows that we can replace $\widetilde{\phi}_y$ by a more convenient function $\widehat{\phi}_y$ measuring the squared distance between $x$ and $y^+ := y - \frac{1}{L}\nabla \phi(y)$, the latter being the point obtained after taking a single GD step from $y$ with learning rate $\frac{1}{L}$, disregarding the constraint:

**Lemma 3.2.** *For any $y \in \mathbb{R}^d_{\leq c}$ we have $M_{V,\widetilde{\phi}_y}(y) = M_{V,\widehat{\phi}_y}(y)$, where*

$$\widehat{\phi}_y(x) := \left\| x - \left( y - \frac{1}{L}\nabla \phi(y) \right) \right\|^2 = \|x - y^+\|^2 = \sum_{i=1}^d \left( x_i - y_i^+ \right)^2. \tag{5}$$

The proof can be found in Appendix A.2. To summarize, the V step of the PV method (Algorithm 1), i.e., $x = M_{V,\phi}(y)$, can be approximated via the "linearized V step"

$$x := M_{V,\phi}(y) \approx M_{V,\widehat{\phi}_y}(y) := \hat{x}. \tag{6}$$

Our next lemma says that the above approximation is in a certain sense natural reasonable provided that $\phi$ is $L$-smooth[5] on $\mathbb{R}^d_{\leq c}$, i.e., provided that

$$\phi(x) \leq \phi(y) + \langle \nabla \phi(y), x - y \rangle + \frac{L}{2} \|x - y\|^2, \qquad \forall x, y \in \mathbb{R}^d_{\leq c}. \tag{7}$$

**Lemma 3.3** (Monotonicity). *Let $y \in \mathbb{R}^d_{\leq c}$. If $\phi$ is $L$-smooth on $\mathbb{R}^d_{\leq c}$, then $\phi\left(M_{V,\phi}(y)\right) \leq \phi(\hat{x}) \leq \phi(y)$, where $\hat{x}$ is the point obtained from $y$ by the linearized V step* (6).

Indeed, the point $\hat{x}$ obtained via the linearized V step can not have a worse loss than the previous point $y$. Of course, one hopes that the loss will strictly decrease so that the method makes progress. From a practical perspective, the key advantage of linearized V step is that it can be performed much faster compared to the vanilla V step. The proof of Lemma 3.3 can be found in Appendix A.3.

Note that since $\widehat{\phi}_y(x)$ is separable (see (8)), each entry/weight of $x$ can be optimized independently of others. For scalar quantization, each individual problem can be solved in $\mathcal{O}(\log_2(c))$ time using binary search in sorted version of $V(y)$. For vector quantization, there are specialized optimization procedures for efficiently minimizing the $L_2$ error (see Appendix D)

**Key challenge.** The main caveat with linearized V step is that it may be impossible to make small gradient-based updates to low-bitwidth discrete weights. More specifically, in (6), one must update the discrete assignments to approximate $y^k - \frac{1}{L}\nabla \phi(y^k)$. However, for low-bit weights, the desired update $\frac{1}{L}\nabla \phi(y^k)$ can be smaller than the lowest possible increment to obtain a quantized vector. As a result, the optimal solution to (6) is often $y^k$ itself. In such a situation, the algorithm will get stuck on $y^k$, which is undesirable. This problem is especially pronounced in deep LLMs, where $L$ can be very large, or, from a practitioner's point of view, where one needs a small learning rate. In practice, as we explore in Section 4.2, the lowest learning rate where the algorithm makes *any* updates at all is already too large for optimization, leading to divergence.

---

[5]It is possible to consider different class of functions instead; e.g., Lipschitz functions. In such a case, we would us a different approximation. For simplicity of exposition, we work with $L$-smooth functions.

| **Algorithm 2** PV-Tuning: Optimization | **Algorithm 3** PV-Tuning: Implementation, one step |
|---|---|
| **Require:** initial parameters $x^0 \in \mathbb{R}^d_c$, objective function $\phi : \mathbb{R}^d \to \mathbb{R}$, subspace size $\tau \in [d]$ | **Require:** quantized `model`, subspace size `tau` |
| 1: **for** $k = 0, \ldots, K - 1$ **do** | 1: `deq_model := dequantize_weights(model)` |
| 2: $\quad \triangleright$ **P step:** update $V(x)$ by backprop | 2: **for** $t = 1, \ldots, T$ **do** |
| 3: $\quad y^k = \underset{y \in \mathbb{R}^d_{\leq c}}{\arg\min}\{\phi(y) \, : \, P(y) \supseteq P(x^k)\}$ | 3: $\quad$ `loss = deq_model(next_batch()).loss` |
| | 4: $\quad$ `loss.backward()` $\triangleright$ accumulate gradients |
| 4: $\quad \triangleright$ **V step:** choose a subspace $\mathcal{S}^k$ & update $P(x)$ | 5: **end for** $\qquad\qquad$ for P and V steps |
| 5: $\quad \mathcal{S}^k = \underset{1 \leq i \leq d}{\arg\,\mathrm{top}\,\tau} \; |\nabla_i \phi(y^k)| \; \triangleright$ find $\tau$ largest | 6: $\triangleright$ **P step:** update codebooks by backprop |
| 6: $\quad \widehat{\phi}_{y,\mathcal{S}^k}(x) := \left\| x - \left( y - \frac{1}{L_{\mathcal{S}^k}} Z^k\left(\nabla\phi(y)\right)\right)\right\|^2$ | 7: `grad_phi = deq_model.weight.grad` |
| | 8: `grad_codebooks = backprop(grad_phi)` |
| 7: $\quad x^{k+1} \underset{=}{\,} \arg\min_x \left\{ \widehat{\phi}_{y^k,\mathcal{S}^k}(x) : V(x) \subseteq V(y^k)\right\}$ | 9: `model.codebooks = adam(grad_codebooks)` |
| | 10: $\triangleright$ **V step:** choose a subspace `s` and update codes |
| 8: **end for** | 11: `update = adam(grad_phi) - deq_model.weight` |
| | 12: `s = choose_subspace(update, tau)` |
| | 13: `model.codes[s] = find_nearest(update[s])` |

Many popular strategies for discrete fine-tuning can be seen as attempts to reconcile coarse low-precision weights with the need to make small updates. These include straight-through estimation, stochastic rounding, or adding regularizers that push the solution to (6) away from $y^k$. We review straight-through estimation in Appendix E.1 and stochastic rounding in Appendix E.2.

### 3.3 Linearized subspace V step

Here we ask the following question: **Can we modify the PV method so as to force the V step to make a larger update?** In other words, we need an optimization algorithm that updates quantized weights either by a sufficiently large increment, or not at all.

A natural example of such an algorithm is coordinate descent (CD) [43, 58], or more generally, subspace descent [26, 38]. Instead of updating all parameters by a small margin, CD in each iteration chooses a single parameter, and makes a large update instead. This strategy can be generalized to updating more parameters at the same time, which leads to subspace descent methods.[6] The parameters to be updated can be chosen either greedily, (e.g., several $i \in [d]$ with the largest magnitude of the partial derivative $|\nabla_i \phi(\cdot)|$), or at random, or through a variety of other means.

Let $\mathcal{S}^k \subset [d]$ be the set of parameters/weights/coordinates we wish to update at iteration $k$. We choose $|\mathcal{S}^k| = \tau \ll d$. Let $Z^k : \mathbb{R}^d \to \mathbb{R}^d$ be the linear mapping defined as follows: $\left(Z^k(x)\right)_i = x_i$ if $i \in \mathcal{S}^k$ and $\left(Z^k(x)\right)_i = 0$ if $i \notin \mathcal{S}^k$. We now formulate the linearized *subspace* V step:

$$x^+ := M_{V, \widehat{\phi}_{y,\mathcal{S}^k}}(y) := \arg\min_{x \in \mathbb{R}^d}\left\{\widehat{\phi}_{y,\mathcal{S}^k}(x) \; : \; V(x) \subseteq V(y)\right\},$$

$$\text{where} \quad \widehat{\phi}_{y,\mathcal{S}^k}(x) := \left\| x - \left( y - \frac{1}{L_{\mathcal{S}^k}} Z^k\left(\nabla\phi(y)\right)\right)\right\|^2, \tag{8}$$

and $L_{\mathcal{S}^k} > 0$ is a smoothness parameter of $\phi$ associated with the subspace spanned by the parameters belonging to $\mathcal{S}^k$. This detail is important because $L_{\mathcal{S}^k} \ll L$ when $\tau \ll d$. *When estimating Lipschitz constants for real LLMs, we found that it is lower by at least one order of magnitude*, making it possible to train with sufficiently large step sizes (see details and $L_{\mathcal{S}^k}$ estimates in Appendix F).

Note that, necessarily, $x_i^+ = y_i$ for $i \notin \mathcal{S}^k$. The remaining $\tau$ entries of $x^+$ can be identified exactly by searching a discrete space of size $|V(y)|^\tau$, which is feasible if $c = \mathcal{O}(1)$ and $\tau = \mathcal{O}(1)$, for example.

In practice, it means that *the algorithm can apply large updates to quantized LLM weights, with the caveat that should only update a fraction of them at a time*. This allows us to perform the linearized V step with sufficiently large "learning rate" to make non-trivial (i.e., $x^{k+1} \neq y^k$) improvements to quantized weights even without straight-through estimation or stochastic rounding.

We formulate the full procedure in Algorithm 2. The algorithm performs the P step by directly optimizing $V(x)$ (i.e., codebooks) by backprop as described in Section 3.1. For the V step, the algorithm greedily chooses a subset of $\tau$ quantized weights for update, then updates them using Eq. (8). The $\arg\,\mathrm{top}\,\tau$ operator finds $\tau$ indices with the largest absolute gradient values and builds a subspace of $\mathbb{R}^d_{\leq c}$ where only these values can be changed, and the rest must be equal to $y^k$.

---

[6]Very closely related methods include block coordinate descent and compressed gradient descent with sparsification operators such as RandK or TopK [4, 7].

### 3.4 Implementation details

To speed up convergence, we use adaptive learning rates for both P and V steps. In Eq. 8, we replace $\nabla\phi(y)$ with a single Adam [37] update, as depicted in Algorithm 3. In preliminary experiments, we found that this results in a significant convergence speedup. When choosing the subspace $\mathcal{S}^k$, we select weights based not on $|\nabla_i\phi(y)|$, but on the magnitude of Adam update for that weight. For simplicity, we greedily choose the $\tau$ weights with the largest update norm within each weight matrix.

This could be further improved through better techniques for choosing $\mathcal{S}^k$ explored in Appendix Q. We also found that, despite the fact that PV-tuning by itself outperforms straight-through estimation, we could achieve slightly better accuracy by combining PV-tuning with straight-through estimation. We explore this in more detail in Section 4.2).

We describe our approach for preparing the calibration data in Appendix G. We found that the preprocessing used in several recent PTQ works introduce a small bias when sampling the calibration data, leading to somewhat worse fine-tuning accuracy. For fairness, we always compare representations (Section 4.1) and algorithms (Section 4.2) using the same pre-processing.

**Fine-tuning efficiency.** The most compute-intensive part of PV tuning is computing the gradients $\nabla\phi(\cdot)$, which is done through repeated forward and backward passes on an LLM. To reduce the number of gradient accumulations, we reuse gradients for P and V steps within one iteration. We use mixed precision, gradient checkpointing and batch accumulation to train more efficiently; for larger LLMs such as LLAMA 3 70B we also use sharding and optimizer offloading (see Appendix H). Our code can train 7B LLMs on a single GPU, while larger ones (e.g. 70B) fit into a single machine with $8\times$A100. In terms of wall-clock time, PV-tuning takes up to $1.5\times$ longer than the fine-tuning procedure of [71] and requires additional memory in order to hold $\nabla\phi(x)$.

## 4 Experiments

### 4.1 Evaluating quantized representations with finetuning

Before evaluating PV-tuning, we need to choose the quantized representation to be fine-tuned. We therefore compare popular weight representations from recent works on LLM quantization (see Section 2). To better isolate the effect of the weight representation, we evaluate them in three configurations: i) when quantizing a single LLM layer, in terms of MSE, ii) full model quantization in terms of perplexity without finetuning and iii) with finetuning.

We compare several recently proposed quantized representations (see details in Appendix J):

1. **GPTQ:** scalar uniform quantization with channel-wise and block-wise scales [22],

2. **SpQR:** an extension of block-wise GPTQ with learned sparse outliers [18],

3. **VQ:** basic vector quantization with a single codebook [72] with multi-step training.

4. **AQLM:** additive vector quantization with multiple learned codebooks [21],

5. **QuIP#**: vector quantization with lattices and incoherence processing [71],

6. **VQ/AQ + outliers:** vector/additive quantization with sparse outliers via pruning [66, 8],

7. **VQ/AQ + lowrank:** vector/additive quantization with Low-Rank Compensation (LoRC) [79],

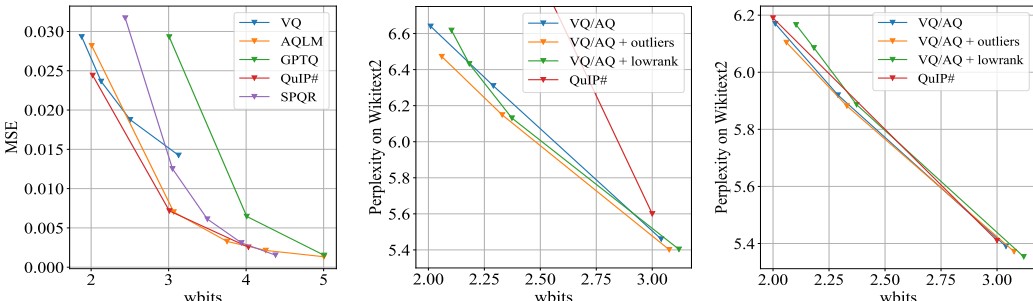

Figure 2: **(left)** L2 errors for 17th layer of LLAMA 2 7B with different representations. Full model perplexity on WikiText-2 is reported without finetuning **(middle)** and with fine-tuning **(right)**.

We run all three experiments on LLAMA 2 7B model [69], calibrating on the RedPajama [13] dataset that best approximates the original pre-training data. When evaluating single layer errors, we report the L2 error in attention query projection outputs of a fixed transformer block, with other blocks exhibiting similar behavior. For full model evaluation, we report quantized model perplexity on WikiText-2 [45] dataset. We use the same data splits and preprocessing as in most recent PTQ works [22, 42, 18, 70, 21, 71], including the biased preprocessing step that we mentioned in Section 3.4. For fine-tuning, we train continuous parameters only, using the approach from [71]. To compare these diverse representations, we evaluate their quantization errors as a function of average number of bits per parameter. To get a diverse set of bits per parameter, we vary the hyperparameters such as wbits, block size, codebook and group size for vector quantization and the rate of outliers.

Figure 2 summarizes our findings. Overall, vector quantization methods (VQ, QuIP# and AQLM) outperform their scalar counterparts. Outliers and low-rank compensation both reduce error, but this improvement comes at the cost of extra bits per parameter. Interestingly, *the improvement from outliers is significantly smaller when both methods have access to fine-tuning*. Likewise, the improvement from using low-rank adapters also diminishes when comparing fine-tuned models, to a point where it no longer justifies the increase in model size. We provide a more detailed breakdown of results and hyperparameter configurations in Appendix J.

Our main takeaway is that for sub 2 bits per parameter, the vector quantization (VQ) representation can achieve near-optimal quantization accuracy, whether or not it uses outliers, LoRC or incoherence processing. Naturally, this does not reduce the value of prior works since they were designed for different scenarios, typically with a higher number of bits per parameter.

## 4.2 Evaluating Fine-tuning Algorithms

Next, we compare different fine-tuning strategies and ablate our PV-tuning protocol. We design our protocol to be representation-agnostic, i.e. compatible with different quantized representations. To showcase this, we pick three methods from the previous section: GPTQ, VQ and AQLM.

These methods differ not only in their weight representations, but also in how they search for the optimal codes. Namely, GPTQ can scale the target weight and round it to nearest 2-bit integer. In turn, VQ quantizes weights as a group and must find the nearest vector from its codebook, and AQLM uses a multi-step beam search procedure to choose the best combination of codes from both codebooks. Our PV-Tuning implementation uses these search algorithms during the subspace linearized V step (`find_nearest` in Alg. 3). We describe the full PV configuration for each method in Appendix K.

We compare PV tuning against several popular fine-tuning regimens found in the literature. Our first baseline is fine-tuning only continuous parameters, e.g., codebooks or input/output embeddings [71, 74]. The second baseline is training with Straight Through Estimation (STE) [75, 77]. We also test stochastic rounding as described in Appendix E.2. Finally, we evaluate PV tuning combined with STE, but otherwise the same configuration. We set the subspace size $\tau$ equal to the number of weights such that the update satisfies $\|x^{k+1} - x^k\|/\|x^k\| \leq 0.01$, also known as known as trust ratio [81].

The results in Table 1 show that PV-Tuning consistently finds better quantized models, with STE coming consistently second. We explore this further by combining subspace updates with STE, which leads to slightly better perplexity and accuracy in most (but not all) setups.

Table 1: Comparing different fine-tuning strategies for VQ, GPTQ and AQLM on LLAMA 2 7B in terms of perplexity on WikiText-2, C4 and average zero-shot accuracy on tasks from Section 4.3.

| Fine-tuning Method | GPTQ 2.14 bit/w | | | VQ, 1.58 bit/w | | | AQLM, 2.01 bit/w | | |
|---|---|---|---|---|---|---|---|---|---|
| | Wiki2↓ | C4↓ | Acc.↑ | Wiki2↓ | C4↓ | Acc.↑ | Wiki2↓ | C4↓ | Acc.↑ |
| Calibration only (no global fine-tuning) | 3290 | 4125 | 29.0 | 20.26 | 20.09 | 43.42 | 7.38 | 9.34 | 53.2 |
| Continuous params only [71, 21] | 16.77 | 17.53 | 46.27 | 8.17 | 10.99 | 52.14 | 6.69 | 8.77 | 56.57 |
| Naive Linearized PV (no subspace) | 16.73 | 17.48 | 47.68 | 8.19 | 10.94 | 52.08 | 6.68 | 8.75 | 56.51 |
| Stochastic Rounding [53] (tuned) | 11.97 | 13.07 | 49.79 | 8.02 | 10.64 | 52.31 | 6.56 | 8.39 | 56.68 |
| Straight Through Estimation [77] | 8.79 | 11.04 | 50.61 | 7.76 | 10.26 | 52.58 | 6.41 | 8.63 | 57.04 |
| Subspace Linearized PV (ours, $\tau$=0.01) | 8.49 | **10.78** | **52.17** | 7.38 | 9.47 | 53.36 | 6.13 | 8.35 | 57.81 |
| Subspace Linearized PV+STE ($\tau$=0.01) | **8.43** | 10.82 | 51.90 | **7.32** | **9.35** | **55.22** | **5.90** | **7.43** | **58.19** |

**PV-Tuning over QuIP#** In addition to these three configurations, we also apply PV-tuning to QuIP# [71] — a modification of vector quantization that applies Randomized Hadamard Transform (RHT) before quantization and uses fixed lattices instead of learned codebooks. We experiment with Llama-2 7B model quantized with QuIP# to 2 bits per weight and found that it is possible to significantly improve the model through PV-Tuning. For instance, PV-tuning improves WikiText-2 perplexity from 6.19 (QuIP# with built-in continuous fine-tuning) to 5.71 (PV-Tuning + STE). Since original 16-bit model has a perplexity of 5.13, this corresponds to almost halving the quantization error in terms of perplexity. We report additional details for QuIP# with PV-Tuning and full evaluation results in Appendix L and include it to Table 2 as "QuIP#+PV".

**On the choice of hyperparameters for 1-bit vector quantization.** There are several possible hyperparameter configurations for vector quantization (VQ) that fall into 1-1.1 bit range. One can either use larger codebooks for longer groups (vectors), or smaller codebooks for shorter groups accordingly. In our main evaluations, we quantized vectors of 16 consecutive weights with 14-16 bit codebooks to fit into the desired bitwidth. However, we later found that it is more advantageous to choose smaller groups as well as codebooks. We found that 8-bit code per 8 weights outperforms 14-bit code per 16 weights despite having near-identical bitwidth (due to smaller codebooks). We report this configuration as "PV (gs8)" in Table 2 and provide additional experiments in Appendix M.

### 4.3 Large-scale Evaluation & Discussion

Finally, we evaluate the resulting PV algorithm with a vector quantization backbone and KL objective on a range of popular LLM models. For this section, our goal is to evaluate our approach holistically for different models and target bit-widths, comparing against the best known baselines in common settings. To that end, we evaluate on LLAMA 2 & 3 [69], MISTRAL 7B [34] and PHI-3 Mini-4k-Instruct [1] at 1–2.5 bits per parameter (averaged over all transformer layers).

We report perplexity on WikiText-2 [45] and C4 [54] validation sets, zero-shot accuracy on Wino-Grande [60], PiQA [67], HellaSwag [83], ARC-easy and ARC-challenge [12] via the LM Eval Harness [24]. We follow the exact evaluation setup from GPTQ [22]. We compare against QuIP [70], BiLLM [32], PB-LLM [62], DB-LLM [10], AQLM [21], OneBit [77], QuIP# [71], the latter three using fine-tuning. For LLAMA 3, we use baselines from [33] and re-evaluate perplexity in our setup.

Table 2: Quantized model perplexity on **WikiText-2**↓ [45] & **C4**↓ [54] and the **Average**↑ accuracy on 5 zero-shot tasks [24] for various models and bitwidths. Arrows ↑ / ↓ mean higher / lower is better.

| Size | Method | Avg bits | Wiki2↓ | C4↓ | Average↑ | Size | Method | Avg bits | Wiki2↓ | C4↓ | Average↑ |
|---|---|---|---|---|---|---|---|---|---|---|---|
| | | | LLAMA 2 model family | | | | | | LLAMA 3 model family | | |
| 7B | – | 16 | 5.12 | 6.63 | 64.80 | 8B | – | 16 | 5.54 | 7.10 | 68.61 |
| | BiLLM | 1.08 | 32.48 | 40.52 | 41.68 | | BiLLM | 1.1 | 28.8 | 257 | 37.90 |
| | OneBit | 1.01 | 9.73 | 11.11 | 50.06 | | PB-LLM | 1.7 | 35.68 | 197.56 | 36.00 |
| | PV-Tuning | 1.02 | 8.28 | 10.37 | 50.66 | | PV-Tuning | 1.01 | **11.17** | **11.67** | **50.01** |
| | PV (gs8) | 1.00 | **7.62** | **9.73** | **53.77** | | QuIP | 2.01 | 76.95 | 98.47 | 36.8 |
| | AQLM | 2.02 | 6.64 | 8.56 | 56.47 | | PB-LLM | 2.00 | 21.74 | 61.04 | 38.80 |
| | QuIP# | 2.01 | 6.19 | 8.16 | 57.51 | | DB-LLM | 2.01 | 12.77 | 14.82 | 51.8 |
| | DB-LLM | 2.01 | 7.23 | 9.62 | 55.12 | | PV-Tuning | 2.01 | **6.99** | **8.29** | **64.36** |
| | PV-Tuning | 2.02 | 5.84 | 7.62 | 61.35 | 70B | – | 16 | 2.59 | 5.78 | 75.37 |
| | QuIP#+PV | 2.01 | **5.71** | **7.51** | **61.81** | | BiLLM | 1.1 | 15.26 | 65.07 | 44.2 |
| 13B | – | 16 | 4.57 | 6.05 | 67.82 | | PV-Tuning | 1.01 | **8.67** | **9.68** | **51.47** |
| | AQLM | 1.97 | 5.65 | 7.51 | 60.59 | | QuIP | 2.00 | 11.63 | 18.54 | 48.71 |
| | QuIP# | 2.01 | 5.35 | 7.20 | 61.45 | | PB-LLM | 2.00 | 10.33 | 28.89 | 46.04 |
| | DB-LLM | 2.01 | 6.19 | 8.38 | 59.41 | | PV-Tuning | 2.07 | **4.57** | **6.56** | **70.38** |
| | PV-Tuning | 1.97 | **5.12** | **6.83** | **64.92** | | MISTRAL 7B v0.1 (A) and PHI 3 Mini-4k-Instruct (B) | | | | |
| | PV-Tuning | 2.19 | **5.05** | **6.74** | **66.05** | 7B (A) | – | 16 | 4.78 | 5.71 | 69.38 |
| 70B | – | 16 | 3.12 | 4.97 | 72.40 | | QuIP# | 2.01 | 6.02 | 6.84 | 62.20 |
| | AQLM | 2.07 | 3.94 | 5.72 | 68.75 | | PV-Tuning | 2.01 | **5.29** | **6.17** | **66.32** |
| | QuIP# | 2.01 | 3.91 | 5.71 | 68.60 | 3.8B (B) | – | 16 | 5.83 | 9.35 | 70.5 |
| | DB-LLM | 2.01 | 4.64 | 6.77 | 65.83 | | AQLM | 2.03 | 8.85 | 12.19 | 60.4 |
| | PV-Tuning | 2.07 | **3.78** | **5.56** | **70.72** | | PV-Tuning | 2.03 | **6.88** | **10.08** | **65.70** |
| | PV-Tuning | 1.14 | 5.52 | 7.50 | 64.58 | | | | | | |

Table 2 summarizes our findings: **PV-tuning with vector quantization outperforms all known methods for 1- and 2-bit per weight**. The closest competitors on LLAMA 2 are QuIP#, AQLM and OneBit, all of which use fine-tuning. The improvements on LLAMA 3 are also remarkable as this model is notoriously hard to compress [33]. We report additional evaluations in Appendix N.

**Pareto-optimality.** A key practical question concerns obtaining optimal quality for the target model size, where a smaller model compressed to 3-4 bits often dominates a larger model compressed to 1-bit. The best known Pareto-optimal bit-width for Llama 2 is 2.5 [21]: compressing a larger model to less than 2.5 bits per weight is inferior to a smaller model quantized to the same total number of bytes. From this perspective, **PV-tuning pushes the Pareto-optimal frontier for LLAMA 2 to 2.0 bits**. This is easiest to see in Table 12: a 2-bit 13B model outperforms any 7B quantization and is comparable with the 16-bit 7B model. The same holds for the 2-bit 70B model.

**Fine-tuning efficiency.** One limitation of our algorithm is that it requires more compute and memory during the fine-tuning procedure. The 7B models can be fine-tuned on a single GPU, our 70B runs require a server with $8 \times A100$ or rely on RAM offloading. PV-Tuning shares this drawback with prior methods based on STE [21, 71], as both methods need gradients w.r.t. dequantized weights. Our longest training run took 2 days on 8 GPUs to outperform all baselines and 8 days to fully converge.

**Inference speed.** PV-Tuning does not change the underlying compressed representation, allowing us to reuse existing high-performance inference kernels. Specifically, VQ+PV can reuse efficient kernels from [21, 71], while GPTQ+PV can use ExLlamaV2 kernels [14]. We report these inference speed evaluations in Appendix O. From a practitioner's point of view, PV-Tuning can significantly improve the accuracy of extreme (1-2 bit) quantized models, making it possible to deploy large LLMs on resource-constrained devices. As a proof of concept, we developed specialized inference engines for running vector-quantized models with PV-Tuning on mobile devices[7] or in the browser[8].

# 5 Conclusions

**Limitations.** We focused our effort on evaluating PV-Tuning with multiple setups and models, but spent relatively little effort tuning our algorithm for each specific setup. For instance, we always use constant learning rate and $\tau$ with no schedule, and always train on the same data. While this shows robustness of PV-Tuning, it also means that our results may be improved with better hyperparameters. For instance, Appendix M shows how PV-Tuning with 1-bit vector quantization can be improved by choosing smaller vector sizes, while Appendix L suggests that PV-Tuning can dramatically improve models quantized with QuIP# and may similarly be applied to other quantized representations. Furthermore, the algorithm could achieve better accuracy by simply training longer and on more data.

**Future work.** This work opens several new research directions. The first is about how to choose $\mathcal{S}^k$: while we found that a greedy strategy works in practice, there may be fundamentally better ways. Another direction is applying PV-Tuning to other quantization niches: our evaluation focuses on extreme weight-only quantization, but the proposed algorithm can be used in weight + activation setting or KV cache quantization. Overall, PV-Tuning shows how an insight from optimization theory can improve LLM quantization and we are excited to see how this develops in future research.

# Acknowledgements

Authors would like to thank Vage Egiazarian, Andrei Panferov and Ruslan Svirschevski for their help and advice on AQLM codebase and running large-scale experiments. We also thank Philip Zmushko and Artem Fedorov for helpful discussions during the early stages of our research. The research of Kai Yi, Konstantin Burlachenko, and Peter Richtárik reported in this publication was supported by funding from King Abdullah University of Science and Technology (KAUST) – Center of Excellence for Generative AI, under award number 5940. We would also like to thank our NeurIPS reviewers for their helpful suggestions, we specifically highlight p3Lv's suggestions to consider smaller codebook sizes and evaluate PV-Tuning with QuIP#, both of which produced interesting findings. Finally, we thank the open-source contributors from llama.cpp[9] and the LocalLlama[10] community for discussions and inspirations on practical use cases of quantized language models, and in particular, Yalda Shabanzadeh and Arthur Aardvark for their help with improving the codebase.

---

[7] https://x.com/black_samorez/status/1821933255744016866

[8] https://galqiwi.github.io/aqlm-rs/about.html

[9] https://github.com/ggerganov/llama.cpp

[10] https://reddit.com/r/LocalLaMA

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

# Appendix

## Contents

# A Proofs

## A.1 Proof of Theorem 3.1

**Part (i):** First, by assumption, we know that $x^0 \in \mathbb{R}^d_{\leq c}$. Assume that $x^k \in \mathbb{R}^d_{\leq c}$ for some $k \geq 0$. Since $P(y^k) \sqsupseteq P(x^k)$, this implies that $y^k \in \mathbb{R}^d_{\leq c}$. Next, since $V(x^{k+1}) \subseteq V(y^k)$, we conclude that $x^{k+1} \in \mathbb{R}^d_{\leq c}$. The claim now follows by induction.

**Part (ii):** Since

$$y^k = \arg\min_{y \in \mathbb{R}^d} \{\phi(y) \; : \; P(y) \sqsupseteq P(x^k)\}$$

and because $y = x^k$ satisfies the constraint $P(y) \sqsupseteq P(x^k)$, we conclude that $\phi(y^k) \leq \phi(x^k)$. Further, since

$$x^{k+1} = \arg\min_{x \in \mathbb{R}^d} \{\phi(x) \; : \; V(x) \subseteq V(y^k)\}$$

and because $x = y^k$ satisfies the constraint $V(y) \subseteq V(y^k)$, we conclude that $\phi(x^{k+1}) \leq \phi(y^k)$. In summary,

$$\phi(x^{k+1}) \leq \phi(y^k) \leq \phi(x^k).$$

**Part (iii):** In view of part (ii), the sequence $\{\phi(x^k)\}_{k=0}^{\infty}$ is non-increasing. By assumption, it is bounded below. Hence, it converges to its infimum:

$$\lim_{k \to \infty} \phi(x^k) = \inf_{k \in \{0,1,\dots\}} \phi(x^k).$$

## A.2 Proof of Lemma 3.2

$$
\begin{aligned}
M_{V,\widetilde{\phi}_y}(y) \;\; &:= \;\; \arg\min_{x \in \mathbb{R}^d_{\leq c}} \left\{ \widetilde{\phi}_y(x) \; : \; V(x) \subseteq V(y) \right\} \\
&= \;\; \arg\min_{x \in \mathbb{R}^d_{\leq c}} \left\{ \phi(y) + \langle \nabla \phi(y), x - y \rangle + \frac{L}{2}\|x - y\|^2 \; : \; V(x) \subseteq V(y) \right\} \\
&= \;\; \arg\min_{x \in \mathbb{R}^d_{\leq c}} \left\{ \langle \nabla \phi(y), x \rangle + \frac{L}{2}\|x - y\|^2 \; : \; V(x) \subseteq V(y) \right\} \\
&= \;\; \arg\min_{x \in \mathbb{R}^d_{\leq c}} \left\{ 2\left\langle \frac{1}{L} \nabla \phi(y), x \right\rangle + \|x - y\|^2 \; : \; V(x) \subseteq V(y) \right\} \\
&= \;\; \arg\min_{x \in \mathbb{R}^d_{\leq c}} \left\{ \left\| x - \left( y - \frac{1}{L}\nabla\phi(y) \right) \right\|^2 \; : \; V(x) \subseteq V(y) \right\} \\
&= \;\; \arg\min_{x \in \mathbb{R}^d_{\leq c}} \left\{ \widehat{\phi}_y(x) \; : \; V(x) \subseteq V(y) \right\} \\
&= \;\; M_{V,\widehat{\phi}_y}(y).
\end{aligned}
$$

## A.3 Proof of Lemma 3.3

First, note that

$$
\begin{aligned}
\phi(\hat{x}) \quad &\overset{(8)}{=} \quad \phi\left( M_{V,\widehat{\phi}_y}(y) \right) \\
&\overset{\text{Lemma 3.2}}{=} \quad \phi\left( M_{V,\widetilde{\phi}_y}(y) \right) \\
&\overset{(3)}{=} \quad \min_{x \in \mathbb{R}^d} \{ \widetilde{\phi}_y(x) \; : \; V(x) \subseteq V(y) \} \\
&\overset{(4)}{=} \quad \min_{x \in \mathbb{R}^d} \left\{ \phi(y) + \langle \nabla \phi(y), x - y \rangle + \frac{L}{2}\|x - y\|^2 \; : \; V(x) \subseteq V(y) \right\}. \quad (9)
\end{aligned}
$$

Since $y \in \mathbb{R}_{\leq c}^d$, any $x \in \mathbb{R}^d$ satisfying $V(x) \subseteq V(y)$ must also satisfy $x \in \mathbb{R}_{\leq c}^d$. So, in view of $L$-smoothness of $\phi$ on $\mathbb{R}_{\leq c}^d$, we can bound the last expression in (9) from below via

$$\phi(\hat{x}) \overset{(9)+(7)}{\geq} \min_{x \in \mathbb{R}^d} \{\phi(x) : V(x) \subseteq V(y)\}$$
$$\overset{(3)}{=} \phi\left(M_{V,\phi}(y)\right).$$

Finally, since $x = y$ satisfies the constraint $V(x) \subseteq V(y)$, we can upper bound the same expression via

$$\phi(\hat{x}) \overset{(9)}{\leq} \phi(y) + \langle \nabla\phi(y), y - y \rangle + \frac{L}{2}\|y - y\|^2$$
$$= \phi(y).$$

# B   Approximate PV Algorithm

We now introduce the pseudocode of an approximate PV meta-algorithm; the idea is to replace the P and V steps with some approximate computations to be defined later.

---

**Algorithm 4** Approximate PV Algorithm

---

1: **Parameters:** starting point $x^0 \in \mathbb{R}_{\leq c}^d$
2: **for** $k = 0, 1, \ldots$ **do**
3:     $y^k \approx M_{P,\phi}(x^k)$
4:     $x^{k+1} \approx M_{V,\phi}(y^k)$ (for example, we can use the method from Section B.1 or the method from Section B.2)
5: **end for**

---

Next, we describe two new approximations of the V step.

## B.1   Approximate V step, variant 1 (non-accelerated)

We now describe an algorithm computing an approximation to $M_{V,\phi}(y)$:

1. Start with some $y \in \mathbb{R}_c^d$ and choose sufficiently large $L > 0$, number of iterations $T$
2. Set $z^0 = y$
3. For $t = 0, \ldots, T - 1$ iterate:
    (i) Define $\widehat{\phi}_{z^t}(\cdot) := \left\| \cdot - \left(z^t - \frac{1}{L}\nabla\phi(z^t)\right) \right\|^2$
    (ii) Set $z^{t+1} = M_{V,\widehat{\phi}_{z^t}}(z^t)$
4. Output: $z^T$

The method is constructed so that $z^T \approx M_{V,\phi}(y)$. If use this subroutine with $T = 1$ to approximate the V step in the PV method, we recover what we earlier called the linearized PV method. Choosing sufficiently large $T \geq 2$ may be advantageous.

## B.2   Approximate V step, variant 2 (accelerated)

We now describe a different algorithm for computing an approximation to $M_{V,\phi}(y)$:

1. Start with some $y \in \mathbb{R}_c^d$ and choose sufficiently large $L > 0$, number of iterations $T$
2. Choose a suitable decreasing sequence of positive scalars $\{\alpha_t\}$, with $\alpha_0 = 1$ and $\lim_{t\to\infty} \alpha_t = 0$
3. Set $z^0 = y$
4. For $t = 0, \ldots, T - 1$ iterate:

(i) Define $\widehat{\phi}_{z^t}(\cdot) := \left\| \cdot - \left( z^t - \frac{1}{L} \nabla \phi(z^t) \right) \right\|^2$

(ii) Set $z^{t+1} = (1 - \alpha_t) \, M_{V,\widehat{\phi}_{z^t}}(z^t) + \alpha_t z^0$

5. Output: $z^T$

The method is constructed so that $z^T \approx M_{V,\phi}(y)$. This approach is based on Halpern acceleration of fixed point methods [51], and as such, may be sometimes effective.

## C   Generalization to Other Quantization Algorithms

In Section 3.1, we define $\mathbb{R}^d_{\leq c}$ as a set of vectors with at most $c$ unique items. This translates to the idea of k-means quantization, a scalar nonlinear quantization where each weight is rounded to one of $c$ centroids found by clustering the weights. Below, we show how this can be generalized to linear quantization, vector quantization, additive quantization, and others.

Linear quantization is the most basic and widely used type of quantization where **weights are stored as integers**, possibly multiplied by a scale and added to a zero point. The simplest way to account for this quantization type is to declare that weight values are integers up to $c$: $V(x) = (0, 1, 2, ..., c - 1)$. After that, one can treat scales / zero points as an extra non-quantized parameter, similar to biases or layer-normalized scales. This extra parameter interacts with weights by multiplication or addition, and hence it can be updated by backprop, similarly to other non-quantized weights. Equivalently, once can declare that $V(x) = (0, s, 2s, ..., s \cdot (c - 1))$ for arbitrary $s \in \mathcal{R}$. Both options lead to equivalent fine-tuning algorithms where the V step does not change and the P step has an additional condition.

Next, let us discuss vector quantization. Consider a quantization that splits $x$ into 2-dimensional groups (non-intersecting pairs of adjacent weights) and encodes these weights as one of $c$ 2-dimensional codes that form its codebook. This can be viewed as two sets of weights (odd and even) quantized with scalar quantization, except that values in the two sets have the same partitioning $P(x)$. In other words, if two values in the odd half belong to the same partition, the corresponding values in the other half also belong to the same partition, though the values themselves can be different. Alternatively, one can simply write down a version of $\mathbb{R}^d_{\leq c}$, where $V(\cdot)$ is a set of 2-dimensional vectors, not scalars. Likewise, higher-dimensional vector quantization translates to higher-dimensional items in $V(\cdot)$.

Both the P and V steps for vector-quantized weights follow the same general principle: P-step can be approximated by backprop with slightly more trainable parameters. In turn, the V step can be done by trying all values in V(x) and selecting the one with the lowest $\widehat{\phi}(\cdot)$. A more compute-efficient version of the V step for this case is described in Appendix D.

RVQ and Additive Quantization can be treated as learning two separate sets of vector-quantized parameters. However, a more efficient way would be to run the V step to find the best combination of codes via beam search [5].

Quantization with sparse outliers [17, 18, 42] can be seen as learning two distinct matrices with different definitions of $\mathbb{R}^d_c$: one is quantized and the other is sparse (for outliers). Similarly, quantized weights with low-rank adapters (e.g. [28]) can treat the adapter as an additional non-quantized parameter for the P step. This makes PV-tuning potentially extensible to neural network pruning.

## D   Efficient Linearized V Step for Vector Quantization

As we describe in Section 3.2, the linearized V step minimizes the squared error between quantized weights and the updated "target" weights. Here, we explain how one can compute and minimize the squared error efficiently in practice. To simplify the notation for this section, we define the objective as $\|x - B\|^2$ where $B$ is the target vector set by the linearized V step. Following the definition of squared $L2$ norm, this objective can be re-written as follows:

$$\|x - B\|^2 = \|x\|^2 - 2\langle x, B \rangle + \|B\|^2. \tag{10}$$

Consider the first term: $\|x\|^2 = \sum_{i=1}^{d} x_i^2$ . Since $x \in \mathbb{R}_{\leq c}^d$, this term is a sum of at most $c$ unique terms. Abusing your notation, this can be rewritten as

$$\|x\|^2 = \sum_i^c V_i(x)^2 \cdot |P_i(x)|,$$

where $V_i(x)$ is $i$-th unique element in $x$ and $|P_i(x)|$ is the number of such elements.

The second term is also a sum of $c$ unique values:

$$-2 \cdot \langle x, B \rangle = -2 \sum_{i=1}^{d} x_i B_i = -2 \sum_{i=1}^{c} \left[ V_i(x) \cdot \sum_{i \in P_i(x)} B_i \right].$$

The third term does not depend on $x$.

If you know the objective for some given $x$, you can efficiently compute $\phi$ for all neighboring $\hat{x} \in \mathcal{N}_1(x)$ where you only change one index. For the sake of formality, let us define the set of such neighboring $\hat{x}$ as follows:

$$\mathcal{N}_1(x) := \{\hat{x} \in \mathbb{R}_{\leq c}^d : V(\hat{x}) = V(x), \|x - \hat{x}\|_0 = 1\} \tag{11}$$

Consider one $\hat{x} \in \mathcal{N}_1(x)$ where only $k$-th value changed (i.e. $x_k \neq \hat{x}_k$). Then,

$$\phi(\hat{x}) - \phi(x) = \|\hat{x}\|^2 - \|x\|^2 - 2 \cdot \langle \hat{x} - x, B \rangle + \|B\|^2 - \|B\|^2 \tag{12}$$

$$\phi(\hat{x}) - \phi(x) = \hat{x}_k^2 - x_k^2 - 2 \cdot (\hat{x}_k - x_k) \cdot B_k + 0 \tag{13}$$

Note that, for any $\forall \hat{x} \in \mathcal{N}_1(x)$, there are $c^2$ possible values for $\hat{x}_k^2 - x_k^2$ and another $c^2$ unique values for $2 \cdot (\hat{x}_k - x_k) \cdot B_k$, regardless of $d$, since there are $c$ unique values in both $v$ and $\hat{x}$. This allows for an efficient local search algorithm:

1. Let $x^0$ be the input to $M_P$

2. Compute and save all $2c^2$ possible red and blue values

3. Compute $\phi_0 = \phi(x^0)$

4. for t = 0, ...:

5.      for $\hat{x} \in \mathcal{N}_1(x^t)$:

6.         find $k : \hat{x}_k \neq x_k^t$ (there's only one such $k$)

7.         compute $\phi(\hat{x}) = \phi(x^t) + \hat{x}_k^2 - x_k^2 - 2 \cdot (\hat{x}_k - x_k) \cdot B_k$

8.      $x^{t+1} := \underset{\hat{x} \in \mathcal{N}_1(x^t)}{\arg\min} \phi(\hat{x})$       (minimum from array of pre-computed values)

In practice, this can be extended from greedy (local) search to semi-greedy beam search. These practical algorithms are described in AQ, LSQ, and LSQ++. Algorithms for $\|A(x - B)\|^2$ are explained in AQLM and probably other works.

# E   Gradient-based Strategies for Training Quantized Models

In this section, we overview possible solution to the general problem of training / fine-tuning neural networks with quantized weights. We focus on strategies that train by gradient descent with additional measures to deal with coarse-grained weights.

In principle, there are also gradient-free methods for quantized training, such as Evolution Strategies [56], Bayesian Optimization [48] and others. However, these gradient-free methods have so far not gained popularity for large language model quantization. Adapting these methods to the scale and dimensionality of LLMs would likely require extra effort. Thus, we leave these methods outside the scope of our work and focus on gradient-based optimization.

**Reminder: gradient-based training of quantized models.** We describe the general framework for training quantized weights in Sections 3.1 and 3.2. To summarize, the training algorithm computes the gradient w.r.t. de-quantized weights as though they were continuous, then uses these gradients to update continuous (P step) and discrete (V step) parameters. The core problem with this approach is that, when discrete parameters are very coarse (e.g. low-bit quantization), gradient updates are no longer large enough to make *any* changes and are lost to the "rounding error". We review two strategies for circumventing this problem: straight-through estimation and stochastic rounding.

## E.1 Straight-through Gradient Estimation

Straight-through gradient estimation is a technique for training neural networks with discrete components that ignores these discrete components during backpropagation. Its usage goes back to the Perceptron introduced by Rosenblatt [59]. There, the artificial neuron uses a step function as activation, but the training procedure treats this function as though it was identity. Subsequent works introduce variations of this idea, extend it to multi-layer networks [31], discuss its convergence properties [76, 23, 80]. Other works use straight-through estimation or similar techniques to training neural network with quantized weights [30, 64, 65].

**STE for LLM quantization.** As we discuss in Section 2, straight-through estimation introduces an auxiliary non-quantized weight tensor that is updated using the gradients $\nabla \phi(y)$ w.r.t. quantized weights. The quantized weights are then updated to best approximate this auxiliary buffer, usually in terms of $L2$ error. As a result, if an update to $y - \frac{1}{L}\nabla\phi(y)$ is not large enough to change the parameter, it is still accumulated in a straight-through "buffer". Eventually, the cumulative effect of several such updates will be large enough that $y^k$ will no longer be the solution to Equation (6).

This strategy prevents the algorithm from stalling, but it does so at the cost of convergence guarantees [80]. When applied to extreme LLM quantization (Section 4.2, straight-through estimation initially improves $y^k$, but then stops improving and oscillates. We also tried several a variant of straight-through estimation [65] that introduce stochasticity to forward pass. When applied to extreme LLM quantization, this variant did not diverge like naive STE, but trained much slower and did not reach the same optimum as "deterministic" STE. We attribute this to the fact that adding noise during training can slow down convergence, which also applies to stochastic rounding (Appendix E.2).

## E.2 Stochastic Rounding

Stochastic (or probabilistic) rounding [82, 87, 2, 29] is one of the techniques that can circumvent stalling when training low-precision weights. To recall, the linearized V step (6) can be seen as rounding $y^+ := y - \frac{1}{L}\nabla\phi(y)$ to the nearest quantized weight in $\mathbb{R}_c^d$, which often happens to be $y$ itself. To circumvent the problem of rounding back to $y$, one can instead round stochastically, to one of the two adjacent values that $y^+$ falls between. Let's denote these two adjacent values $x_l$ and $x_r$ for left and right. The probability of rounding is inversely proportional to the rounding error (distance), or, in terms of the objective,

$$p(\text{round to } x_l) = \frac{\widehat{\phi}(x_l)^{-1/2}}{\widehat{\phi}(x_l)^{-1/2} + \widehat{\phi}(x_r)^{-1/2}}.$$

This way,

$$\underset{p(\text{round to } x)}{E} x = y - \frac{1}{L}\nabla\phi(y).$$

The main drawback of stochastic rounding is t introduces noise, it changes the underlying optimization problem. Intuitively, if the optimal $x^\star$ is adjacent to a significantly worse solution, the method may oscillate between rounding to either side. This rounding noise increases further as we consider lower quantization width. In Section 4.2 we exploit his phenomenon for real-world LLMs and find that stochastic rounding converges find significantly worse solutions, presumably because at every step, some portion of LLM weights will be rounded poorly. On top of that, when used for vector quantization, stochastic rounding is either intractable or biased.

**Stochastic rounding for vector quantization.** To recall, stochastic rounding for non-vector quantization needs to find two quantized values: the nearest neighbor above the solution, and the nearest neighbor below it. It will then round to either of the two values inversely proportionally to their rounding errors.

However, this intuition no longer works if you consider more complex quantization schemes such as vector quantization, additive quantization, quantized low-rank adapters, and others. In vector quantization, a group of weights is encoded jointly as one vector from a fixed set (usually called codebook or lattice). For simplicity, let us consider the case where the weight group size equals 2, i.e. weights are quantized in pairs.

For a pair of two eights, we can no longer rely on the fact that they have one neighbor from above and one from below. Instead, they may have any number of adjacent "clusters" they can be rounded to. Intuitively, a pair of weights is a point in 2-dimensional that can have neighbors from left, right, top, bottom, and any diagonals. Formally, to determine a full list of neighbors, we can run Delaunay triangulation on all vectors from the codebook (or lattice) plus the target vector that needs to be rounded, then find all points that share a triangle with the target vector.

Unfortunately, this procedure can be very expensive, especially for higher-dimensional vectors. A popular practical approximation to stochastic rounding for vector quantizations is to find K (e.g. 2) nearest vectors from the codebook, then use the probability formula from scalar stochastic rounding:

$$p(\text{round to } x_i) = \widehat{\phi}(x_i)^{-1/2}/(\sum_{j}^{K} \widehat{\phi}(x_j)^{-1/2})$$

However, unlike the original stochastic rounding, this approximation does not guarantee that

$$\underset{p(\text{round to } x_i)}{E} x_i = y - \frac{1}{L}\nabla\phi(y). \tag{14}$$

For a (counter)example, if there is a high density of codes on one side of the target, all K (e.g. 2) nearest neighbors will be on the same side. As a result, this approximate stochastic rounding is biased and further changes the optimization result.

**Stochastic rounding with temperature.** When used for LLM quantization, the main problem with stochastic rounding is that it introduces noise to the training procedure. This is important because modern LLMs [69, 68, 1] typically train **without** dropout or similar noise layers. This is because the dataset is huge and the training suffers not from overfitting, but from not fitting the data enough.

Training with stochastic rounding makes optimization inherently noisy as if using dropout, making it harder to train. What is worse, low-bitwidth models produce more noise than high-bitwidth due to larger intervals between quantized values. This additional noise makes it difficult for the model to fit the training data tightly. For extreme 1-bit training, we often observed that the training objective (cross-entropy) would increase instead of decreasing due to sheer amount of rounding noise.

To combat this issue, we introduce stochastic rounding with temperature $\tau$ (hyperparameter):

$$p(\text{round to } x_l) = \frac{\widehat{\phi}(x_l)^{-1/(2\tau)}}{\widehat{\phi}(x_l)^{-1/(2\tau)} + \widehat{\phi}(x_r)^{-1/(2\tau)}}.$$

Setting $\tau < 1$ results the algorithm keeping the original weights more often, which reduces the training noise at the cost of slower updates. In Table 1 (Section 4.2), we try $\tau \in \{1, 0.5, 0.1, 0.01\}$ and choose the best result for each setup that involves stochastic rounding. It can be shown to converge as long as $\tau$ is annealed, but it changes the underlying optimization problem similarly to dropout. In principle, it is possible to gradually anneal $\tau$ during training to make the algorithm unbiased in the limit.

### E.3   Comparing Discrete Optimization Techniques

Finally, we can compare the two above strategies and our proposed approach from Section 3.3.

**From the optimization perspective,** the most popular variant of straight-through estimation is known to *not* converge to a stable solution. While, in practice, STE can still significantly improve model quality (see Table 1), it is still a heuristic. In turn, stochastic rounding can be seen as an additional source of noise to stochastic gradient descent which can converge to a stable solution when $\tau$ is

gradually reduced to 0. In contrast, subspace PV does not introduce noise or instability and does not require annealing.

The most notable difference of PV-Tuning from STE is that the former tries to update all weights on every step, whereas our algorithm only updates a chosen subset. We give an explanation on why updating all weights is problematic at the end of Section 3.2.

**From the efficiency perspective,** training with straight-through estimation requires storing an additional set of buffers on device memory to accumulate weight updates. Unlike the quantized weights, these buffers need to be stored in higher precision (half or full) to accumulate smaller updates that would be lost on quantized buffers. As a result, straight-through estimation requires additional memory for fine-tuning. Stochastic rounding and subspace PV (w/o STE) do not need these buffers, but they still need to accumulate high precision gradients w.r.t. de-quantized weights and store optimizer statistics for those weights in higher precision. To summarize, all methods require significantly more memory than naive (P-only) fine-tuning, but straight-through estimation has higher memory overhead.

As for the computational overhead, both STE, stochastic rounding and subspace PV introduce additional computations and therefore increase step complexity. Of the three alternatives, the subspace PV algorithm is slightly faster since it only runs the discrete optimization on a small portion (subspace) of model weights per step, while stochastic rounding has somewhat higher overhead due to the complicated rounding procedure, especially for vector quantization. However, **this overhead is small in practice: most of the LLM training time is spent accumulating the gradients on a large training batch, which is not affected by any of the three algorithms.** In principle, it should be possible to reduce the compute / memory overhead both with technical improvements and better optimization algorithms, but we leave this investigation to future work.

## F  On $L$-smoothness of LLM Objective in Sparse Subspaces

The classical definition of $L$-smoothness for differentiable function $f : \mathbb{R}^d \to \mathbb{R}$ represented by requirement

$$\|\nabla f(x) - \nabla f(y)\| \leq L\|x - y\|, \qquad \forall x, y \in \mathbb{R}^d.$$

If function $f(x)$ is twice continuously differentiable, then it is easy to show that function $f$ is $L$-smooth if and only if $\|\nabla^2 f(x)\| \leq L$. If striving to find the minimum value of $L$ then via following the definition, one has to select $L$ as $L = \max_{x \in \mathbb{R}^d} \left( \|\nabla^2 f(x)\| \right)$.

If the domain of function $f(x)$ is restricted to any subset $S \subseteq \mathbb{R}^d$, then the global $L$ smooth constant can only decrease for a new function. It can be observed from the fact that $\forall S \subseteq \mathbb{R}^d$ we have

$$\|\nabla^2 f(x)\| = \max_{v \in \mathbb{R}^d \backslash 0} \left( \nabla^2 f(x) \cdot v/\|v\| \right) \geq \max_{v \in S \backslash 0} \left( \nabla^2 f(x) \cdot v/\|v\| \right), \forall x \in \mathbb{R}^d.$$

Sparse sub-spaces satisfy this requirement; therefore, this theoretical observation is valid in this circumstance.

Another observation is that the definition of $L$ smooth constant has a global notion. However, for practical purposes, for the first-order training algorithm what matters is $L$-smoothness constant for the function $f(x)$ with the domain restricted to the trajectory of iterates only. Unfortunately, the training process iterates follow the prior unknown path in $\mathbb{R}^d$ space.

Below we demonstrate two approximate schemas for evaluating $L$-smoothness constant for a function with a domain (artificially) restricted to the trajectory induced by iterates generated by Gradient Descent (GD) for functions $f(x)$ with different subspace sizes and in general with different optimization trajectories, but with the same start iterate (model). We have performed 10 iterations of GD for training auto-regressive Llama-like `LLama-160M` [46] and `TinyLlama-1.1B` [85] models using automatic tokenizer from these models. We trained all linear layers in these models. The used step size for GD is $10^{-4}$.

**Schema I: Estimating $L$ along the trajectory of iterates without capturing local curvature.**

After running GD for $10$ iterations the $L$ smooth constant has been estimated along trajectory $s = \{x_1, x_2, \ldots, x_{10}\}$ with approximated as

$$\hat{L} := \max_{x_i, x_j \in z, x_i \neq x_i} \left( \frac{\|\nabla f(x_i) - \nabla f(x_j)\|}{\|x_i - x_j\|} \right).$$

Results are presented in Tables 3, 4. From them, we can see that $\hat{L}$ estimate along the trajectory of iterates have the same property as global $L$, namely during restricting subspace of training variables the $L$-the smooth constant is non-increasing, and in practice substantially decreasing. This schema exploits available information on gradient oracles in iterates in $s$ and iterates $s$ itself. This schema represents an estimation of upper bound $\hat{L}$ on the true value of $L$.

Table 3: Estimated $L$ along the GD trajectory for `LLama-160m` (Schema I).

| Subspace Size | Number of Trainable Parameters | Estimated $\hat{L}$ |
|---|---|---|
| 5% | 2.36M | 10.01 |
| 10% | 8.26M | 14.40 |
| 20% | 17.69M | 305.72 |
| 30% | 24.77M | 498.16 |
| 40% | 36.57M | 919.82 |
| 60% | 60.75M | 5454.79 |
| 70% | 85.52M | 6915.06 |
| 85% | 102.04M | 7043.19 |
| 100% | 113.25M | 7251.50 |

Table 4: Estimated $L$ along the GD trajectory for `TinyLlama-1.1B` (Schema I).

| Subspace Size | Number of Trainable Parameters | Estimated $\hat{L}$ |
|---|---|---|
| 5% | 13.11M | 33.24 |
| 10% | 49.28M | 143.00 |
| 20% | 136.84M | 2159.22 |
| 30% | 242.75M | 2369.63 |
| 40% | 369.10M | 2638.11 |
| 60% | 582.48M | 5185.92 |
| 70% | 684.20M | 5901.73 |
| 85% | 831.52M | 6091.04 |
| 100% | 968.88M | 9480.57 |

**Schema II: Estimating $L$ along the sequence of iterates with capturing local curvature.**

The previous schema used a fixed sequence of iterates $s = \{x_1, x_2, \ldots, x_{10}\}$ essentially estimate the $L$-smoothness constant along the piece-wise linear interpolated path along $s$. In the next schema, we approximate $L$-smoothness constant as

$$\hat{L} = \max_{x_i \in s} \left( \|\nabla^2 f(x_i)\| \right).$$

Therefore this schema exploits a sequence of points $s$ and selects the maximum in absolute values eigenvalue for all matrices $\nabla^2 f(x_i)$. Computing any spectral information for a matrix with big dimensions can be challenging.

The approximate schema that we have used to compute $\|\nabla^2 f(x)\|$ leverages several observable facts. First, $\|\nabla^2 f(x)\| = \max(|\lambda_i(\nabla^2 f(x))|)$, where $\lambda_i$ is $i - th$ eigenvalue for $\nabla^2 f(x)$. Second, to identify the maximum eigenvalue in the absolute value we can use the normalized Power Iteration algorithm [47], which requires execution only the hessian-vector product. Third, we can use Taylor expansion and forward difference approximation for $\nabla f(x + r)$:

$$\nabla f(x + r) - \nabla f(x) = \nabla^2 f(x) \cdot r + \mathcal{O}\left(\|r\|^2\right)$$

The $K$ approximate hessian-vector product can be accomplished with $K + 1$ gradient oracle calls. For the experiment, we run *Power Iteration* for a prior known number of iterations equal to 10. In fact *Power Iteration* does not converge in case of degeneracy such as a situation when the matrix has two maximum eigenvalues in absolute values but with opposite signs, and the convergence rate is determined by the absolute value of the ratio of the second-largest-magnitude eigenvalue to the first. We ignore these aspects in our heuristic approximate Algorithm 5.

---

**Algorithm 5** Approximate Matrix-free Algorithm for Computing $\|\nabla f(x)\|$

---

1: **Parameters:** Point $x \in \mathbb{R}^d$, fixed $\gamma \in \mathbb{R}$ such as $\gamma = 10^{-5} \cdot x$ for numerical stability.
2: $r^0 \sim_{\text{u.a.r}} \mathbb{R}^d$
3: $g = \nabla f(x)$
4: **for** $k = 0, 1, \ldots, K$ **do**
5: $\quad \hat{r^k} = r^k / \|r^k\|$
6: $\quad r^{k+1} = 1/\gamma \left( \nabla f(x + \gamma \hat{r^k}) - g \right)$ // Approximate computation of $r^{k+1} \approx \nabla^2(f) \cdot \hat{r^k}$.
7: **end for**
8: **Output:** Approximate eigenvector $r^{K+1} / \|r^{K+1}\|$ corresponding to $|\lambda_{\max}| \approx \|r^{K+1}\|$.

---

Results are presented in Tables 5, 6. From them, we can see that also this notion of $\hat{L}$ estimate along the set of iterates has the same property as global $L$, namely during restricting subspace of training variables the $L$-the smooth constant is non-increasing similar to previous estimation method.

Table 5: Estimated $L$ along the GD iterates for `LLama-160m` with local curvature (Schema II).

| Subspace Size | Number of Trainable Parameters | Estimated $\hat{L}$ |
|:---:|:---:|:---:|
| 5% | 2.36M | 10.96 |
| 10% | 8.26M | 791.30 |
| 20% | 17.69M | 878.37 |
| 30% | 24.77M | 1202.00 |
| 40% | 36.57M | 1918.04 |
| 60% | 60.75M | 5262.77 |
| 70% | 85.52M | 5325.83 |
| 85% | 102.04M | 5901.21 |
| 100% | 113.25M | 11522.45 |

Table 6: Estimated $L$ along the GD iterates for `TinyLlama-1.1B` with local curvature (Schema II).

| Subspace Size | Number of Trainable Parameters | Estimated $\hat{L}$ |
|:---:|:---:|:---:|
| 5% | 13.11M | 40.30 |
| 10% | 49.28M | 146.93 |
| 20% | 136.84M | 4366.38 |
| 30% | 242.75M | 4487.38 |
| 40% | 369.10M | 6767.58 |
| 60% | 582.48M | 8983.85 |
| 70% | 684.20M | 15445.54 |
| 85% | 831.52M | 21167.06 |
| 100% | 968.88M | 28629.24 |

## G Calibration Data Matters

For a fair comparison, we run our algorithm using the same calibration data as the baseline algorithms, typically a sample from RedPajama [13]. However, the way we handle this calibration data differs from most baselines [21, 18, 71].

When analyzing their codebase, we found that these algorithms resample calibration data by taking a random excerpt of a fixed length from a random document in the calibration data, both sampled uniformly. However, with this approach, the data from longer documents (e.g. books) are underrepresented compared to shorter ones (e.g. news articles), which biases the calibration data.

Upon further investigation, we believe that new methods blindly copied this code from each other, going back to GPTQ [22] and possibly further. This choice was harmless for GPTQ since it requires relatively little calibration data; however, full model fine-tuning like in QuIP# [9] and AQLM [21], works better on unbiased data.

To remove the bias, we use standard[11] LM preprocessing that concatenates all documents, then splits them into evenly sized chunks that become training sequences. The benefits from debiasing range from insignificant to as large as 0.15 perplexity for some models. To compensate for that, we run experiments with the same preprocessing protocol unless explicitly stated otherwise.

# H  Additional Engineering Considerations

When done naively, the longest operation is the discrete update (8) that runs discrete optimization on all LLM parameters. For scalar quantization, this step does simple rounding and runs nearly instantly. In turn, applying it to vector quantization requires solving a discrete optimization algorithm (e.g. beam search) for every group of weights. However, since equation (8) can only update weights within $S^k$, we can skip discrete optimization for any weight that was not among the chosen few. As a result, when training models with up to 70 billion parameters, we search less than one billion times per step.

The next longest operation is computing the gradients $\nabla\phi(\cdot)$, needed both for P and V steps. This involves running LLM multiple forward and backward passes on batches of texts and accumulating the gradients. To reduce the overhead from gradient computation, we compute the gradients once using mixed precision, then reuse these gradients for one P and one V step, respectively. In other words, we switch from alternating P and V steps to performing these steps simultaneously. We also reuse these gradients to update any parameters not affected by quantization: input embeddings, normalization layers, biases, etc.

To limit VRAM usage, we use gradient checkpointing [27], batch accumulation. For larger models, we also use parameter sharding[12] [55] and optimizer offloading [57]. We need these techniques so that smaller 7B LLMs can be trained on a single GPU and larger ones with 70B parameters fit into a single machine with 8×A100. Still, PV-tuning takes up to 1.5× longer than tuning continuous parameters and uses more memory (during training) to hold the gradients w.r.t. dequantized weights.

---

[11]from e.g. `https://github.com/huggingface/transformers/blob/main/examples/pytorch/language-modeling/run_clm.py`

[12]We use PyTorch FullyShardedDataParallel [52, 40] and wrap discrete weights as in `bitsandbytes` [19]

# I    Additional evaluations of perplexity

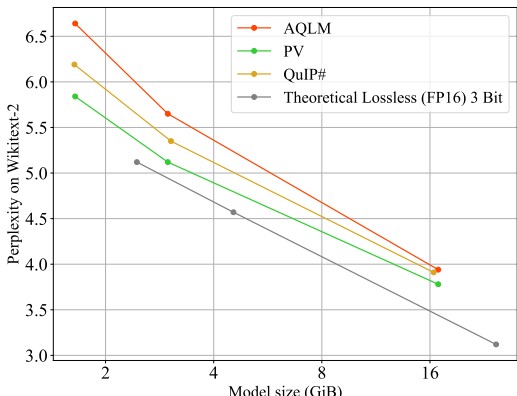

Figure 3: WikiText-2 perplexity of 2-bit quantized LLAMA 2 models as a function of model size (GiB) compared to a theoretical lossless 3 bit compressed model (i.e. float16 perplexity numbers paired with 3-bit model sizes).

# J    Additional Details for Section 4.1

Here, we describe some of the implementation details we used to optimize different quantized representations. For every optimizations, we check that this optimization improves the algorithm in both MSE and perplexity and does not require additional resources that would result in unfair comparison.

**Vector Quantziation.** The original algorithm quantizes all weights a single pass over input channels. We found that it works slightly better if we make multiple such passes and, between passes, update codes by Adam to minimize the same objective [72]. This is resembles QuIP# with no RHT & lattices or AQLM with no additive quantization. For simplicity, we also use a single shared codebook (e.g. instead of groupwise codebooks).

**VQ+outliers** To select outlier coordinates, we use `https://github.com/fmfi-compbio/admm-pruning` that outperforms the SpQR outlier criterion [18] in both L2 error and perplexity (when both criteria are used alongside vector quantization). We re-run the ADMM procedure multiple times during optimization, resulting in an EM-like algorithm.

**VQ+lowrank.** We experimented with two different initializations for low-rank correction: a) quantizing weight matrix, then training LoRC on quantization error, as in [79] and b) initializing LoRC to approximate the reference weight matrix, the applying vector quantization to LoRC errors. Of these two approaches, we found that the latter one produces a (slightly) better solution in both MSE and perplexity.

**Additional representation evaluations.** Below, we report some additional quantized representation configurations that extend our evaluations from Section 4.1. For convenience, we report them both as per-method perplexity values in Table 7 and the combined plots in Figure 4.

Table 7: Comparison of WikiText-2 Perplexity for each method with and without fine-tuning for quantizing Llama 2 7B model. For each method, **PPL no FT** denotes its perplexity without fine-tuning, whereas **PPL w/ FT** is perplexity with fine-tuning. We use the same setup as in Section 4.1.

| Method | Avg bits | PPL no FT↓ | PPL w/ FT↓ | Method | Avg bits | PPL no FT↓ | PPL w/ FT↓ |
|--------|----------|------------|------------|--------|----------|------------|------------|
| –      | 16       | 5.12       | –          | GPTQ   | 2        | 3290       | 16.77      |
| VQ/AQ  | 2.01     | 6.64       | 6.17       | GPTQ   | 3        | 8.52       | 7.26       |
| VQ/AQ  | 2.29     | 6.31       | 5.92       | GPTQ   | 4        | 5.87       | 5.74       |
| VQ/AQ  | 3.04     | 5.46       | 5.39       | SpQR   | 2.09     | 12.19      | 9.90       |
| QuIP#  | 2.00     | 8.22       | 6.19       | SpQR   | 3.45     | 5.48       | 5.37       |
| QuIP#  | 3.00     | 5.60       | 5.41       | SpQR   | 3.98     | 5.29       | 5.21       |

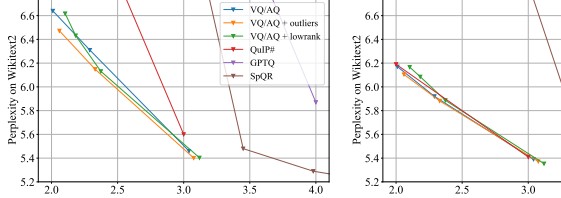

Figure 4: Llama2 7B perplexity on WikiText-2 after compression without fine-tuning **(left)** and with fine-tuning **(middle)**. On the **(right)**, there is a plot that combines the first two, allowing for a better comparison of each method with and without fine-tuning. **Compression algorithms without fine-tuning are represented with dashed lines, while algorithms with fine-tuning are represented with continuous lines.**

Table 8: Evaluation of quantized LLAMA 2 models for 2.x bits per weight. We use the same setup as in the main paper (Section 4.2 with an extra baseline). As requested, we finetune the model in 16-bit precision for the same number of steps, then quantize it with AQLM, reported as **"FT+AQLM"**. Finally, the "Finetuned" row corresponds to an uncompressed 16-bit model finetuned without quantization. We hypothesize that fine-tuning the model has little effect since we train on a dataset resembling its original pretraining data.

| Size | Method | Avg bits | Wiki2↓ | C4↓ | ArcC↑ | ArcE↑ | HellaSwag↑ | PiQA↑ | WinoGrande↑ | Average↑ |
|------|--------|----------|--------|-----|-------|-------|------------|-------|-------------|----------|
| 7B   | –      | 16.00    | 5.12   | 6.63 | 43.43 | 76.3  | 57.14      | 78.07 | 69.06       | 64.80    |
|      | AQLM   | 2.02     | 6.64   | 8.56 | 33.28 | 61.87 | 49.49      | 73.56 | 64.17       | 56.47    |
|      | PV-Tuning | 2.02  | 5.84   | 7.62 | 38.40 | 71.17 | 53.50      | 76.99 | 66.69       | 61.35    |
|      | Finetuned | 16    | 5.13   | 6.59 | 43.46 | 76.47 | 56.96      | 78.14 | 68.91       | 64.79    |
|      | FT + AQLM | 2.02  | 6.48   | 8.36 | 33.29 | 62.45 | 49.31      | 73.49 | 64.82       | 56.67    |

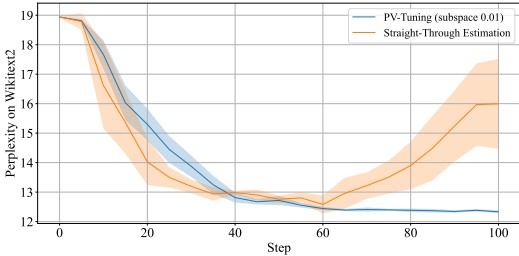

Figure 5: Learning curve for PV-tuning and STE algorithms, when tuning tinyllama model with 2x8g8 AQLM quantization (2 codebooks with 8 bits per code and input groupsize equal to 8).

# K    Additional Details for Section 4.2

**VQ:** vector quantization as a simple near-optimal algorithm. We use a single 16-bit codebook with group size 16 (over input dimension) and per-channel trainable scales over output dimension. During P step, we update the codebook, scales and non-quantized model layers by backprop. During V step, we try every code in the codebook and choose the one that minimizes (6).

**GPTQ:** scalar quantization with block-wise scales and zero points. We use 2-bit base codes and block size 128. During P step, we update the scales and non-quantized parameters by backprop. In turn, V step performs simple rounding to nearest (scaled) integer.[13]

**AQLM:** we perform scalar quantization with two 8-bit codebooks and group size 8 and channel-wise steps. This was originally published in [21] as the "speed-optimized" configuration capable of fast lookup-based inference. During P step, we update both codebooks, as well as scales an non-quantized parameters by backprop. On V step, we run beam search with beam size 8 with gradient-updated dequantized weight as target.

**Training.** We minimize Kullback–Leibler divergence as our loss function for all three representations. More specifically, we fine-tune the quantized "student" model to approximate the predictions (logits) of a "teacher" — the same model prior to quantization. We fine-tune on the same RedPajama sample as in calibration. More specifically, we use the official one-billion-token sample[14] provided by the dataset authors [13]. We use a batch size of $2^{20}$ ($\approx$ 1M) tokens, split into batches of model-specific sequence length (e.g. 4096 tokens for LLAMA 2, 8192 for LLAMA 3). In early experiments, we found PV to be resilient to the choice of batch size, consistently training with up to $4\times$ smaller batches.

**Hyperparameter tuning:** we tune the hyperparameters for each method individually. For all algorithms, we tune learning rate on a logarithmic scale out of (1e-4, 3e-4, 1e-3, 3e-3, 1e-2). For methods involving discrete optimization, we tune learning rate for P and V step separately. The optimal configuration for STE and stochastic rounding is to use learning rate 3e-4 for both codes and codebooks. The optimal configuration for subspace linearized PV and the same with STE is to use learning rate 3e-4 for P step and 3e-3 for codes. Curiously, the subspace methods are stable even with larger step sizes for codes, e.g. 1e-2, whereas unrestricted methods (e.g. pure STE) are not.

For stochastic rounding, we found that the unbiased rounding [53] causes the model quality to quickly degrade, likely due to the fact that the algorithm makes too many weight changes due to rounding. The results we reported in Table 1 use stochastic rounding with temperature 0.2. In other words, we measure the distances to 2 nearest bins and round to each bin proportionally to distance$^{-5}$. We also tried gradually annealing the rounding temperature to 0, but achieved only insignificant improvements in accuracy and perplexity (<0.01). To simplify evaluation, we do not use annealing in Table1.

For PV-tuning and PV-tuning with STE, we always set $\tau$ to maximum number of weights such that updating them satisfies $||x^{k+1} - x^k||/||x^k|| < 0.01$. We implement this by trying to update large portions of weights (in our implementation, we update 0.01 of all weights at a time) until the total change exceeds the constraint. Once it does, we rollback weights in the last chunk to $x^k$ until the constraint is satisfied. As an implementation quirk, we always allow at least one weight to change even if changing just one weight already exceeds $0.01 \cdot ||x^k||$. However, we didn't see this in practice.

We also found that Lamb[15] [81] is more stable when training with large batch sizes, but converges to approximately the same accuracy. We use $\beta_1$=0.9 and $\beta_2$=0.95, same as in most LLM training configurations [69, 86, 61]. *We do not use learning rate decay for simplicity.* It is likely possibly to improve our results by annealing the learning rates during training or using a warmup. We intentionally avoid this to reduce the number of "moving parts" and simplify evaluation. Overall, we found that PV-tuning is about as sensitive to hyperparameters as continuous-only LLM fine-tuning [71, 21, 63].

In the codebase, we release several additional PV-tuned models with a more careful choice of training hyperparameters and calibration datasets: for instance, when quantizing instruction tuned models, we found that calibrating the model on chat-like data results in better accuracy.

---

[13]We also found that we can perform V step by running the entire OPTQ calibration while using $y - \frac{1}{L}\nabla\phi(y)$ as target weights, showing the flexibility of the general PV framework. This variant converges to the same values, but is much slower due to having to re-accumulate the L2 error hessians for each V step.

[14]https://huggingface.co/datasets/togethercomputer/RedPajama-Data-1T-Sample

[15]Lamb is a variant of Adam that limits the norm of weight updates relative to the norm of weights.

## L  PV-Tuning of QuIP#

QuIP# is a popular modification of classical vector quantization for neural network compression. Unlike standard VQ, QuIP# does not quantize weights directly, but instead applies a Randomized Hadamard Transform (RHT) and quantizes the projected weight representations. This is done by applying the Hadamard transform to the rows and columns of the $m \times n$ matrix and multiplying them by random vectors $S_V \sim \mathcal{U}\{\pm 1\}^n, \quad S_U \sim \mathcal{U}\{\pm 1\}^m$ after each transformation. These vectors do not need to be stored, but may be generated on the fly from a random seed.

The main objective of this transformation is to ensure that the Hadamard-transformed weight matrix adheres to a normal distribution. With this in mind, QuIP# no longer needs to learn the optimal quantization "codebook", but may instead use the optimal lattice for quantizing normal distribution. As a result, QuIP# does not need to store the codebooks in memory, thereby reducing its overall bitwidth. For instance, for Llama-2 7B vector quantization with group size 8 and 16-bit codes, VQ produces a 2.29 bit model while QuIP# has slightly less than 2.01 bits per parameter.

We apply PV-Tuning to QuIP# in a manner similar to Vector Quantization. We start with a model already quantized using the original QuIP# algorithm, specifically employing the official 2-bit quantization of the Llama 2 7B model[16]. We then optimize this model using the same hyperparameters as those for Vector Quantization (see Appendix K).

During P step, we optimize quantization scales (SU, SV), as well as any non-quantized model parameters (embeddings, LM head, normalization scales and biases), but not the codebook, since QuIP# relies on a predefined lattice structure that cannot be trained directly. During V step, we update the discrete codes responsible for choosing a vector out of the said lattice using the same procedure as in standard vector quantization.

We compare PV-Tuning against several other fine-tuning strategies: no fine-tuning, the built-in fine-tuning procedure proposed in [71], as well as the technically improved version of that procedure using the observations described in in G (denoted as "improved FT"). We report the resulting perplexities and zero-shot accuracies in Table 9 and include them in other relevant tables as "QuIP#+PV".

Table 9: Evaluation of LLAMA-2 7B quantized using QuIP# with various fine-tuning strategies. We report perplexity on WikiText-2 [45] & C4 [54] and zero-shot accuracy. The **Average** is the mean accuracy of 5 zero-shot tasks. Primary metrics are Wiki2 (PPL), C4 (PPL) and Average (Accuracy).

| Method | Avg bits | Wiki2↓ | C4↓ | ArcC↑ | ArcE↑ | HellaSwag↑ | PiQA↑ | WinoGrande↑ | Average↑ |
|---|---|---|---|---|---|---|---|---|---|
| – | 16.00 | 5.12 | 6.63 | 43.43 | 76.3 | 57.14 | 78.07 | 69.06 | 64.80 |
| QuIP# (w/o FT) | 2.01 | 8.22 | 11.01 | 28.84 | 55.56 | 42.94 | 71.38 | 62.43 | 52.23 |
| QuIP# (built-in FT) | 2.01 | 6.19 | 8.16 | 34.60 | 64.60 | 48.34 | 75.10 | 64.90 | 57.51 |
| QuIP# (improved FT) | 2.01 | 5.92 | 7.82 | 37.63 | 72.39 | 53.07 | 76.12 | 65.35 | 60.91 |
| QuIP# + PV | 2.01 | 5.71 | 7.51 | 39.33 | 72.01 | 54.10 | 77.20 | 66.38 | 61.81 |

In general, we found that QuIP# shows similar if not better improvements from PV-Tuning as VQ and AQLM. Furthermore, when compared to only fine-tuning continuous parameters, QuIP# shows better relative improvement from V steps than traditional vector quantization. We attribute this to the fact that VQ can fine-tune its codebook even without PV-Tuning, whereas QuIP# relies on fixed codebooks that cannot be learned. As a result, discrete parameters (codes) take up a larger fraction of the total model size in QuIP#, allowing V steps to make more significant improvements. The detailed instructions for running these experiments can be found in our official implementation[17], in a separate subsection of the common README file.

## M  On 1-bit Vector Quantization Options

As discussed above, there are several ways to achieve the same number of bits per parameter with vector quantization. In this section, we explore the effect of varying the group size: either quantizing

---

[16] https://huggingface.co/relaxml/Llama-2-7b-E8P-2Bit
[17] https://github.com/Vahe1994/AQLM/tree/pv-tuning

groups of 16 weights with larger codebooks, or groups of 8 weights with smaller ones. The results with different group and codebook sizes are reported in Table 10.

Table 10: Evaluation of LLAMA-2 7B quantized using VQ+PV with different group size (GS, also known as vector dimension) and code bits (CB, s.t. codebook size is $2^{CB}$). We report perplexity on WikiText-2 [45] & C4 [54] and zero-shot accuracy. The **Average** is the mean accuracy of 5 zero-shot tasks. Primary metrics are Wiki2 (PPL), C4 (PPL) and Average (Accuracy).

| GS | CB | Avg bits | Wiki2↓ | C4↓ | ArcC↑ | ArcE↑ | HellaSwag↑ | PiQA↑ | WinoGrande↑ | Average↑ |
|---|---|---|---|---|---|---|---|---|---|---|
| – | – | 16.00 | 5.12 | 6.63 | 43.43 | 76.3 | 57.14 | 78.07 | 69.06 | 64.80 |
| 8 | 8 | 1.00 | 7.62 | 9.73 | 28.84 | 61.66 | 44.66 | 72.14 | 61.56 | 53.77 |
| 8 | 9 | 1.13 | 7.15 | 9.20 | 30.55 | 65.11 | 46.68 | 72.63 | 62.04 | 55.40 |
| 16 | 14 | 1.02 | 8.28 | 10.37 | 25.85 | 57.58 | 40.88 | 68.99 | 57.77 | 50.21 |
| 16 | 16 | 1.58 | 7.32 | 9.35 | 29.44 | 64.14 | 46.03 | 73.12 | 63.38 | 55.22 |

Overall, we figure out that smaller groups provide better performance for the same model size. We hypothesize that PV-Tuning is better able to deal with these configurations because with smaller group size, discrete codes constitute a larger fraction of model parameters, which allows the V step of our algorithm to achieve more significant improvements. In the initial version of manuscript we performed all 1-bit experiments with suboptimal group size 16. More recent results for group size 8 are provided under a separate name "PV (gs 8)" in Table 2.

## N    Additional Details and Evaluations for Section 4.3

In this section, we report additional results for LLAMA 2 & 3, MISTRAL and PHI-3 and discuss baselines. In this section, we always evaluate PV-tuning for vector quantization, using 14-16 bits per codebook for a group of 8 or 16 weights, with each combination fitting a particualr niche. For instance, 16 bits per 8 weights is slightly over 2 bits per weight, whereas 14 bits per 16 weights is either at or below 1 bit per weight, depending on the model size.

We use LLAMA 2 models as our main benchmark as they are well studied in the PTQ community. Here, we gather the latest state-of-the-art algorithms at the time of publication and group them according to their target number of bits, roughly 1-1.7 bits per weight (Table 11) and 2-2.5 bits per weight (Table 12).

Both our training runs and almost all baselines use the same sample of RedPajama data from previous sections[18]. The only exception to this is OneBit that uses a corpora of LLM outputs gathered specifically for that paper [77].

The source code for this method was unavailable until very recently. The official link `https://github.com/xuyuzhuang11/OneBit` used to point to an empty repository until the code was released in a commit `https://github.com/xuyuzhuang11/OneBit/commit/380a6aedc3c060993056ff50b79065e893be99ae` on May 10th. Thus, unfortunately, we did not have time to make OneBit compatible with models except LLAMA 2 7B and 13B that were featured in the original paper.

For LLAMA 3, we evaluate PV-tuning of vector quantization against the baselines introduced in [33]. Curiously, their paper seems to compute perplexity differently than our paper. Since our protocol matches with most prior works [22, 18, 21, 71], we chose to re-evaluate the results from [33] with our perplexity code and not the other way around. We calibrate using the official code [19] and reuse published models where available.

## O    Inference Speed with Vector Quantization Kernels

In this section, we demonstrate that PV-Tuning can achieve speedups by using fast inference kernels from the underlying quantized representation. Since our main experiments use vector quantization,

---

[18]`https://huggingface.co/datasets/togethercomputer/RedPajama-Data-1T-Sample`
[19]`https://github.com/Macaronlin/LLaMA3-Quantization`

Table 11: Evaluation of quantized LLAMA 2 models for 1-1.7 bits per weight, grouped by bitwidth. We report perplexity on WikiText-2 [45] & C4 [54] and zero-shot accuracy. The **Average** is the mean accuracy of 5 zero-shot tasks. Primary metrics are Wiki2 (PPL), C4 (PPL) and Average (Accuracy).

| Size | Method | Avg bits | Wiki2↓ | C4↓ | ArcC↑ | ArcE↑ | HellaSwag↑ | PiQA↑ | WinoGrande↑ | Average↑ |
|---|---|---|---|---|---|---|---|---|---|---|
| 7B | – | 16.00 | 5.12 | 6.63 | 43.43 | 76.3 | 57.14 | 78.07 | 69.06 | 64.80 |
| | BiLLM | 1.08 | 32.48 | 40.52 | 24.4 | 36.2 | 34.8 | 60.6 | 52.4 | 41.68 |
| | OneBit | 1.0 | 9.73 | 11.11 | 29.61 | 41.58 | 52.58 | 68.12 | 58.41 | 50.06 |
| | PV-Tuning | 1.02 | 8.28 | 10.37 | 25.85 | 57.58 | 40.88 | 68.99 | 57.77 | 50.21 |
| | PB-LLM | 1.70 | 69.20 | 80.15 | 25.00 | 28.00 | 27.70 | 53.80 | 49.30 | 36.76 |
| | PV-Tuning | 1.58 | 7.32 | 9.35 | 29.44 | 64.14 | 46.03 | 73.12 | 63.38 | 55.22 |
| 13B | – | 16.00 | 4.57 | 6.05 | 48.38 | 79.42 | 60.03 | 79.05 | 72.22 | 67.82 |
| | BiLLM | 1.10 | 16.77 | 27.54 | 21.84 | 46.84 | 30.97 | 60.61 | 56.75 | 43.40 |
| | OneBit | 1.00 | 8.76 | 10.15 | 33.62 | 43.10 | 56.43 | 70.13 | 61.72 | 53.00 |
| | PV-Tuning | 0.97 | 7.23 | 9.31 | 30.8 | 63.09 | 47.03 | 72.25 | 62.35 | 55.10 |
| | PB-LLM | 1.70 | 151.09 | 144.59 | 21.89 | 35.08 | 24.82 | 54.17 | 52.76 | 37.74 |
| | PV-Tuning | 1.37 | 6.65 | 8.72 | 34.04 | 67.38 | 49.14 | 73.94 | 65.51 | 58.00 |
| 70B | – | 16.00 | 3.12 | 4.97 | 54.35 | 82.74 | 64.79 | 82.15 | 77.98 | 72.40 |
| | BiLLM | 1.08 | 8.41 | 15.19 | 38.91 | 67.3 | 45.71 | 69.7 | 67.64 | 57.85 |
| | PV-Tuning | 1.01 | 6.09 | 8.20 | 38.31 | 71.80 | 53.98 | 75.24 | 68.43 | 61.55 |
| | PB-LLM | 1.70 | 28.37 | 32.63 | 39.89 | 49.50 | 36.62 | 61.43 | 62.80 | 50.05 |
| | PV-Tuning | 1.14 | 5.52 | 7.50 | 42.66 | 74.96 | 56.42 | 77.37 | 71.51 | 64.58 |

Table 12: Evaluation of quantized LLAMA 2 models for 2-2.3 bits per weight, grouped by bitwidth. We report perplexity on WikiText-2 [45] & C4 [54] and zero-shot accuracy. The **Average** is the mean accuracy of 5 zero-shot tasks. Primary metrics are Wiki2 (PPL), C4 (PPL) and Average (Accuracy).

| Size | Method | Avg bits | Wiki2↓ | C4↓ | ArcC↑ | ArcE↑ | HellaSwag↑ | PiQA↑ | WinoGrande↑ | Average↑ |
|---|---|---|---|---|---|---|---|---|---|---|
| 7B | – | 16.00 | 5.12 | 6.63 | 43.43 | 76.30 | 57.14 | 78.07 | 69.06 | 64.80 |
| | QuIP# | 2.02 | 8.22 | 11.01 | 34.60 | 64.60 | 48.34 | 75.10 | 64.90 | 57.51 |
| | AQLM | 2.02 | 6.64 | 8.56 | 33.28 | 61.87 | 49.49 | 73.56 | 64.17 | 56.47 |
| | PV-Tuning | 2.02 | 5.84 | 7.62 | 38.40 | 71.17 | 53.50 | 76.99 | 66.69 | 61.35 |
| | AQLM | 2.29 | 6.29 | 8.11 | 34.90 | 66.50 | 50.88 | 74.92 | 65.67 | 58.57 |
| | PV-Tuning | 2.29 | 5.68 | 7.47 | 38.91 | 72.90 | 53.94 | 77.37 | 67.72 | 62.17 |
| 13B | – | 16.00 | 4.57 | 6.05 | 48.38 | 79.42 | 60.03 | 79.05 | 72.22 | 67.82 |
| | QuIP# | 2.01 | 6.06 | 8.07 | 39.50 | 69.30 | 53.44 | 77.30 | 67.70 | 61.45 |
| | AQLM | 1.97 | 5.65 | 7.51 | 37.80 | 69.78 | 53.74 | 76.22 | 65.43 | 60.59 |
| | PV-Tuning | 1.97 | 5.12 | 6.83 | 43.00 | 75.38 | 57.96 | 78.24 | 70.01 | 64.92 |
| | AQLM | 2.19 | 5.41 | 7.21 | 41.98 | 75.04 | 55.49 | 76.99 | 69.53 | 63.81 |
| | PV-Tuning | 2.19 | 5.05 | 6.74 | 45.65 | 77.57 | 58.00 | 78.07 | 70.96 | 66.05 |
| 70B | – | 16.00 | 3.12 | 4.97 | 54.35 | 82.74 | 64.79 | 82.15 | 77.98 | 72.40 |
| | QuIP# | 2.00 | 4.16 | 6.01 | 48.70 | 77.30 | 60.79 | 80.30 | 75.90 | 68.60 |
| | AQLM | 2.07 | 3.94 | 5.72 | 51.96 | 81.44 | 61.46 | 80.25 | 76.64 | 70.35 |
| | PV-Tuning | 2.07 | 3.78 | 5.56 | 51.88 | 81.02 | 63.07 | 80.74 | 76.87 | 70.72 |

we adopt simplified version of AQLM inference kernels with one codebook of size 16 for groups of 8 consecutive weights. This kernel is not written by us: it was added to the official AQLM implementation by an open-source contributor.

We adapt this inference code to our codebase and switch it to using group size 16 to support out 1.1-1.58 bit models. We evaluate inference speeds on a single Nvidia RTX 3090 GPU using transformers with cuda graphs[20].

---

[20]The specific version of each library can be found in 'requirements.txt'

Table 13: Evaluation of quantized LLAMA 3 models for 1-2.3 bits per weight, grouped by bitwidth. We report perplexity on WikiText-2 [45] & C4 [54] and zero-shot accuracy. The **Average** is the mean accuracy of 5 zero-shot tasks. Primary metrics are Wiki2 (PPL), C4 (PPL) and Average (Accuracy).

| Size | Method | Avg bits | Wiki2↓ | C4↓ | ArcC↑ | ArcE↑ | HellaSwag↑ | PiQA↑ | WinoGrande↑ | Average↑ |
|---|---|---|---|---|---|---|---|---|---|---|
| | – | 16.00 | 5.54 | 7.10 | 50.43 | 80.09 | 60.19 | 79.71 | 72.61 | 68.60 |
| | BiLLM | 1.10 | 28.80 | 65.00 | 17.70 | 36.00 | 28.90 | 56.10 | 51.00 | 37.90 |
| | PV-Tuning | 1.01 | 11.13 | 11.63 | 25.43 | 59.09 | 41.01 | 68.26 | 56.27 | 50.01 |
| | PB-LLM | 1.70 | 35.68 | 197.56 | 17.50 | 31.70 | 27.70 | 52.50 | 50.40 | 36.00 |
| | PV-Tuning | 1.54 | 9.43 | 10.26 | 32.68 | 65.78 | 46.66 | 72.63 | 64.40 | 56.43 |
| 8B | QuIP | 2.00 | 76.95 | 98.47 | 21.30 | 29.00 | 29.20 | 52.90 | 51.70 | 36.80 |
| | PB-LLM | 2.00 | 21.74 | 61.04 | 17.20 | 37.80 | 29.80 | 57.00 | 52.50 | 38.80 |
| | DB-LLM | 2.00 | 12.77 | 14.82 | 28.20 | 59.10 | 42.10 | 68.90 | 60.40 | 51.80 |
| | PV-Tuning | 2.00 | 6.99 | 8.29 | 42.75 | 75.84 | 55.52 | 77.75 | 69.93 | 64.36 |
| | PV-Tuning | 2.30 | 6.76 | 8.10 | 42.32 | 75.46 | 56.21 | 78.45 | 71.67 | 64.82 |
| | – | 16.00 | 2.59 | 5.78 | 60.41 | 86.7 | 66.34 | 82.48 | 80.9 | 75.366 |
| | BiLLM | 1.10 | 15.26 | 65.07 | 25.11 | 46.42 | 37.48 | 58.21 | 53.63 | 44.17 |
| | PV-Tuning | 1.00 | 8.67 | 9.68 | 25.51 | 54.34 | 48.71 | 65.56 | 63.22 | 51.47 |
| 70B | PB-LLM | 1.70 | 16.27 | 54.03 | 25.8 | 49.90 | 34.90 | 56.5 | 53.10 | 44.1 |
| | PV-Tuning | 1.14 | 7.76 | 8.93 | 33.28 | 63.89 | 53.39 | 69.86 | 69.61 | 58.01 |
| | QuIP | 2.00 | 11.63 | 18.54 | 26.50 | 48.90 | 40.90 | 65.30 | 61.70 | 48.70 |
| | PB-LLM | 2.00 | 10.33 | 28.89 | 25.10 | 40.60 | 42.70 | 65.20 | 56.40 | 46.00 |
| | PV-Tuning | 2.07 | 4.55 | 6.54 | 50.77 | 80.22 | 63.85 | 79.22 | 78.06 | 70.42 |

Table 14: Evaluation of quantized MISTRAL V0.1 7B (A) and PHI 3-MINI-4K-INSTRUCT 3.8B (B) models for 1-2.3 bits per weight, grouped by bitwidth. We report perplexity on WikiText-2 [45] & C4 [54] and zero-shot accuracy. The **Average** is the mean accuracy of 5 zero-shot tasks. Primary metrics are Wiki2 (PPL), C4 (PPL) and Average (Accuracy).

| Size | Method | Avg bits | Wiki2↓ | C4↓ | ArcC↑ | ArcE↑ | HellaSwag↑ | PiQA↑ | WinoGrande↑ | Average↑ |
|---|---|---|---|---|---|---|---|---|---|---|
| | – | 16.00 | 4.77 | 5.71 | 50.43 | 80.09 | 60.19 | 79.71 | 72.61 | 68.60 |
| | AQLM | 1.01 | 70.88 | 34.67 | 19.11 | 27.36 | 26.3 | 52.99 | 48.38 | 34.83 |
| | PV-Tuning | 1.01 | 7.58 | 8.14 | 27.73 | 60.82 | 44.33 | 69.97 | 60.38 | 52.65 |
| 7B (A) | QuIP# | 2.01 | 6.02 | 6.84 | 39.76 | 72.14 | 52.95 | 76.71 | 69.30 | 62.20 |
| | AQLM | 2.01 | 6.19 | 6.90 | 30.8 | 49.87 | 25.63 | 56.53 | 57.06 | 43.98 |
| | PV-Tuning | 2.01 | 5.29 | 6.17 | 44.20 | 77.36 | 58.21 | 79.05 | 72.77 | 66.32 |
| | AQLM | 2.27 | 5.78 | 6.55 | 42.06 | 75.17 | 55.09 | 76.93 | 70.24 | 63.90 |
| | PV-Tuning | 2.27 | 5.22 | 6.10 | 45.31 | 77.57 | 58.61 | 79.22 | 70.96 | 66.33 |
| | – | 16.00 | 4.77 | 5.71 | 60.41 | 86.7 | 66.34 | 82.48 | 80.90 | 75.37 |
| | AQLM | 1.03 | 102.54 | 85.20 | 18.86 | 28.16 | 27.13 | 53.54 | 49.80 | 35.50 |
| | PV-Tuning | 1.03 | 11.71 | 14.59 | 21.50 | 49.87 | 34.62 | 65.67 | 54.70 | 45.27 |
| | AQLM | 1.60 | 34.36 | 35.16 | 19.62 | 35.10 | 29.71 | 57.24 | 51.7 | 38.67 |
| 3.8B (B) | PV-Tuning | 1.60 | 9.21 | 12.16 | 30.89 | 63.55 | 43.09 | 70.35 | 61.4 | 53.86 |
| | AQLM | 2.03 | 8.85 | 12.19 | 41.04 | 74.49 | 47.25 | 72.36 | 66.93 | 60.41 |
| | PV-Tuning | 2.03 | 6.88 | 10.08 | 46.84 | 78.24 | 53.75 | 78.67 | 71.03 | 65.71 |
| | AQLM | 2.30 | 8.07 | 11.23 | 47.01 | 78.03 | 50.51 | 75.95 | 70.8 | 64.46 |
| | PV-Tuning | 2.30 | 6.63 | 9.89 | 50.51 | 79.5 | 55.32 | 79.49 | 73.01 | 67.57 |

We were able to acheve 47.4 tokens per second for 7B model, 32.8 tokens per second for 13B model and 7.2 tokens per second for LLAMA 2 70B model. Compared to 16-bit inference code, this results in speedups of 14%, 22% and 28% respectively. Note that PV-Tuning does not make the model inherently faster than, for instance, AQLM. Instead, it can achieve better quality with lower bit models, allowing practitioners to run smaller and faster models for the same accuracy target.

# P  The Choice of the Initial Point $x^0$

Algorithm 1 converges from any initial point $x^0 \in \mathbb{R}_c^d$ (3.1), but it might converge to a different final point $\hat{x}(x^0)$. In this section, we discuss two possible variants of instantiating the initial point.

## P.1  Clipping of $x^\star$

**Notation**: $[d] := \{1, \cdots, d\}$ is the set of $d$ distinct natural numbers from 1 to $d$.

Assume we have the vector $x$ with dimensionality $d$ and let $c \in [d]$. Let us define the clipping operator $C(x) : \mathbb{R}^d \to \mathbb{R}_{\leq c}^d$ in the following way:

$$C(x) = \tilde{x} : \quad V(\tilde{x}) \subseteq V(x) \quad \text{and} \quad |V(\tilde{x})| \leq c \tag{15}$$

We can come up with different variants of clipping operators $C(x)$, but in our experiments, we choose $C(x)$ such that $V(\tilde{x})$ consists of the smallest distinct elements from $V(x)$. We will call this clipping operator by $C_-(x)$.
*Example* P.1. If $x = \{1, 1, 3, 5, 9\}$, then having $c = 2$ the clipping operator $C_-(x) = \tilde{x} = \{1, 1, 3, 3, 3\}$. So, $\tilde{x}$ consists of smallest 2 elements from $V(x) = \{1, 3, 5, 9\}$.

## P.2  Random $x^0 \in \mathbb{R}_c^d$

Now let us define the algorithm for generating random points from $\mathbb{R}_c^d$ (Alg. 6).

---

**Algorithm 6** Generation of Random Point $x^0 \in \mathbb{R}_c^d$

---

1: **Parameters:** dimensionality $d$, number of distinct values $c$
2: Generate a set $U$ with $c$ unique elements: $U = \{u_i \in \mathbb{R}^d\} : |U| = c$.
3: Sample $d$ random elements $x_i^0 \in U$: $x^0 = (x_1^0, \cdots, x_d^0)^T$ such that $|V(x^0)| = c$.

---

Different random $x^0 \in \mathbb{R}_c^d$ provide different loss functions. To get more stable and smoothed results we ran the algorithm (1) with different initial points, generated by algorithm (6). We will denote $r$ as the number of runs with different random initialization.

# Q  Small-Scale Experiments and Interpretation

## Q.1  Objective function

Let $c \in [d]$ and consider the problem

$$\min_{x \in \mathbb{R}_{\leq c}^d} \phi(x), \tag{16}$$

where $\phi : \mathbb{R}^d \to \mathbb{R}$, and $\mathbb{R}_{\leq c}^d \subset \mathbb{R}^d$ is the set of all vectors in $\mathbb{R}^d$ whose $d$ entries take at most $c$ distinct values. In other words, the cardinality of the set

$$V(x) := \{x_1, \cdots, x_d\} \tag{17}$$

is at most $c$. For small experiments, we aim to minimize the following objective

$$\phi(x) = \sum_{i=1}^d a_i (x_i - x_i^\star)^2 = (x - x^\star)^T \Lambda (x - x^\star), \tag{18}$$

where $a_i \in \mathbb{R}^d$, $x^\star$ is a unique optimal point: $\nabla \phi(x^\star) = 0$ and

$$\Lambda = \text{diag}(a_1, \cdots, a_d) = \begin{bmatrix} a_1 & \cdots & 0 \\ \vdots & \ddots & 0 \\ 0 & 0 & a_d \end{bmatrix}. \tag{19}$$

We set $a_i = {}^i\!/_d$ for $i \in [d]$. Hence, we have the Lipschitz continuity for (18) with $L = 1$. Note that the Algorithm 1 does not guarantee to converge to $x^\star$. Let $\hat{x}$ denote the vector to which the Algorithm 1 converges.

## Q.2 Tiny-scale experiments ($d = 6$, $c \in [1, 6]$)

We applied PV algorithm to the problem (16) with $\phi(x)$ being (18) (Fig. 6). Number of runs with different random initial points $r = 50$.

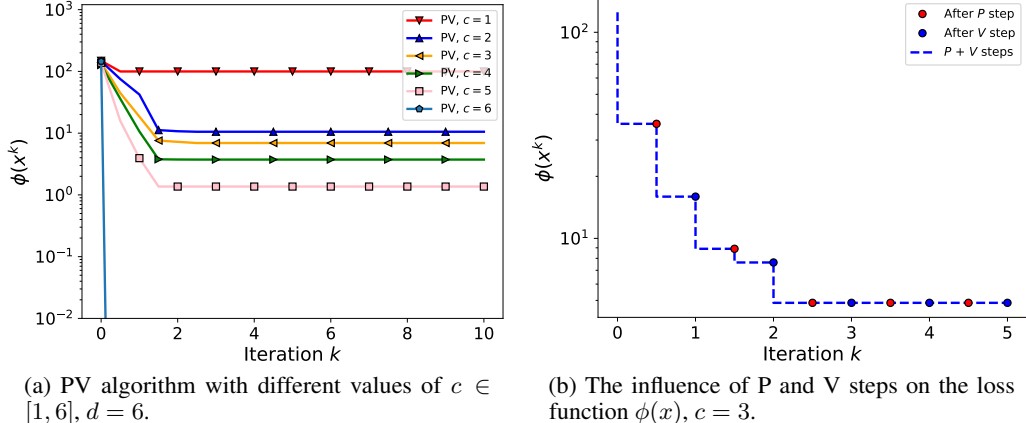

(a) PV algorithm with different values of $c \in [1, 6]$, $d = 6$.

(b) The influence of P and V steps on the loss function $\phi(x)$, $c = 3$.

Figure 6: PV algorithm (1) applied on the very small dimensional ($d = 6$) quadratic objective (18). The starting point $x^0$ is chosen randomly using the ng algorithm (6).

When $c = 1$ we converge just in one step of PV (red line on Fig.6a) because we have only one degree of freedom to play with and we fully utilize it on the first step. One can show that the solution for the case $c = 1$ will be $\hat{x} = \left( \sum_{i=1}^{d} a_i \right)^{-1} \sum_{i=1}^{d} a_i x_i^{\star}$.

Note that as we increase the maximum number of unique elements $c$, the final loss $\phi(\hat{x})$ decreases up until the moment when $c = d$, when the loss is zero (the plot 6a is in logarithmic scale and that is why we cannot see the last line).

We run PV algorithm (1) with different random starting points $x^0 \in \mathbb{R}^d_c$ (6). Different starting points can lead to different local optimum. That is why we ran the algorithm (1) several times with different random initial points $x^0$. A number of runs $r = 50$, we used this value for all further experiments.

Each of the two steps of the PV algorithm contributes to the convergence. To observe this we plotted the loss function over the iterates and explicitly marked the progress of both P and V steps (Fig. 6b). We can see that we have progress during each of these steps and one single P and V step is not enough to obtain a solution even in this very small and simple case.

These simple experiments demonstrate that

1. larger $c$ (smaller compress ratio) provides better final accuracy
2. several P and V steps are needed to converge to the solution

## Q.3 Interpretation of $P(y) \sqsupseteq P(x^k)$

As we mentioned before, we select the initial point $x^0 \in \mathbb{R}^d_c$ randomly such that it has $c$ unique elements in its linear shell $V(x^0)$. To understand the notation $P(y) \sqsupseteq P(x^k)$, defined in the chapter (3.1), let us consider the first step of the algorithm, specifically the P step.

On the first step of the algorithm we select a new point by solving the following optimization problem:

$$y^0 = \arg\min_{y \in \mathbb{R}^d_{\leq c}} \{ \phi(y) \; : \; P(y) \sqsupseteq P(x^0) \}. \tag{20}$$

Here we have a random point $x^0$, with some partition $P(x^0) = \{P_1(x^0), \cdots, P_c(x^0)\} - c$ sets of equal elements from $x^0$. We find $y^0$ such that minimizes $\phi(y^0)$ and have the same or smaller number of partitions: $P(y^0) = \{P_1(y^0), \cdots, P_s(y^0)\}$, where $s \leq c$.

Strict equality of two sets, $P(x^0)$ and $P(y^0)$ would mean that the linear shell – the number of unique elements of the vector $x^0$ is equal to the number of unique elements from $y^0$ ($c = s$). In our PV algorithm we allow $y^0$ to have smaller number of unique elements than $x^0$, so we possibly merge some partitions of $x^0$.

It is worth to mention that in real experiments we observe that the number of unique elements did not change or the change is not significant.

## Q.4  Simple example of $P(y) \sqsupseteq P(x^k)$

Let $d = 8$ and $x = (1, 3, 1, 4, 4, 5, 1, 3)$. Then $x \in R_c^d$, where $c = 4$ since $x$ consists of 4 unique floats: 1, 3, 4 and 5. We have $P_1(x) = \{1, 3, 7\}, P_2(x) = \{2, 8\}, P_3(x) = \{4, 5\}$ and $P_4(x) = \{6\}$. So, $P(x) = \{P_1(x), P_2(x), P_3(x), P_4(x)\} = \{\{1, 3, 7\}, \{2, 8\}, \{4, 5\}, \{6\}\}$. $P(x)$ is a set whose elements are four sets, forming a partition of $\{1, 2, \cdots, 8\}$.

Let $y = \{1, 1, 1, 4, 4, 4, 1, 1\}$. Then $y \in R_c^d$, where $c = 2$. We have $P_1(y) = \{1, 2, 3, 7, 8\}$ and $P_2(y) = \{4, 5, 6\}$. So, $P(y) = \{\{1, 2, 3, 7, 8\}, \{4, 5, 6\}\}$. Notice that each element of $P(x)$ is a subset of some element of $P(y)$. For example, $\{2, 8\}$ is a subset of $\{1, 2, 3, 7, 8\}$ and $\{6\}$ is a subset of $\{4, 5, 6\}$. Because of this, by our definition, $P(y) \sqsupseteq P(x)$.

## Q.5  Small-scale experiments ($d = 100$, $c \in [1, 100]$)

The problem of the algorithm (1) is that the V step requires the full parameter search which gives us the complexity $\mathcal{O}(c^d)$. Even for small tasks, this becomes unpractical to solve.

*Example* Q.1. Let us take $d = 100$ and $c = 10$, then the complexity of one V step will be $\mathcal{O}(10^{100})$. Modern computers can make roughly 10 petaflops or $10^{16}$ calculations per second, hence we will have to wait $\sim 10^{76}$ years to make a single V step.

Let us consider special sets of function $\phi(x)$ that we will call separable functions. This class of functions should satisfy the assumption (Q.2).

**Assumption Q.2** (Separable function)**.** The function $\phi(x) : \mathbb{R}^d \to \mathbb{R}$ can be written in the form $\phi(x) = \sum_{i=1}^d \phi_i(x_i)$, where $\phi_i(\cdot)$ is a mapping $\phi_i(\cdot) : \mathbb{R} \to \mathbb{R}$.

*Example* Q.3.  The objective function (18) is a sum of squares that can be written in the following form:

$$\phi(x) = \sum_{i=1}^d a_i(x_i - x_i^\star)^2 = \sum_{i=1}^d \phi_i(x_i), \tag{21}$$

where $\phi_i(x_i) = a_i(x_i - x_i^\star)^2$. Hence, the objective (18) is a separable function.

One can show that for separable functions (Q.2) the algorithm (1) can be written in the form (7). Hence, for separable functions, we can compute the V step in $\mathcal{O}(c \cdot d)$ operations (instead of $\mathcal{O}(c^d)$), which makes the algorithm (1) practical to use.

---

**Algorithm 7** PV Algorithm with Optimized V step

---

1: **Initialization:** starting point $x^0 \in \mathbb{R}_c^d$
2: **for** $k = 0, 1, \ldots$ **do**
3:      $y^k = \arg\min_{y \in \mathbb{R}^d} \left\{ \phi(y) : P(y) \sqsupseteq P(x^k) \right\}$                  (P step)
4:      **for** $i = 0, 1, \ldots, d$ **do**
5:          $x_i^{k+1} = \arg\min_{x_i \in \mathbb{R}} \left\{ \phi_i(x_i) : x_i \in V(y^k) \right\}$        (Optimized V step)
6:      **end for**
7: **end for**

---

Then we ran the optimized PV algorithm (7) on the quadratic objective (18). The results are presented in (Figure 7). Number of runs with different random initial points $r = 50$.

For these experiments, we see the same results as for tiny scale with $d = 6$ (Q.2): larger $c$ (smaller compress ratio) provides better final accuracy, and several P and V steps have to be done to obtain better solution. In addition, we observe that the PV algorithm does not decrease the dimensionality (number of unique elements) of $V(x)$.

As it was mentioned before, the algorithm (1) is conceptual only and cannot be used in the raw form in practice. That is why we need to move to the experiments with linearized V step.

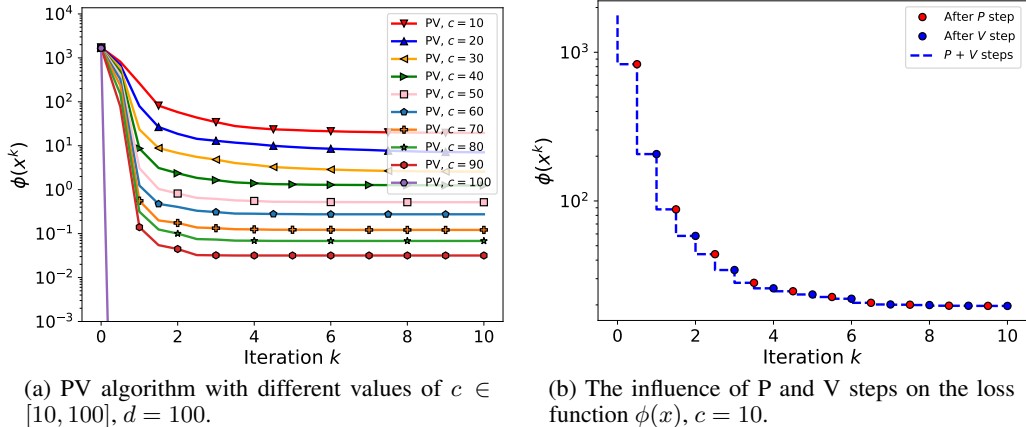

(a) PV algorithm with different values of $c \in [10, 100]$, $d = 100$.

(b) The influence of P and V steps on the loss function $\phi(x)$, $c = 10$.

Figure 7: Optimized PV algorithm (7) applied on the quadratic objective (18), $d = 100$. Number of runs with different random initial points $r = 50$.

## Q.6  Linearized PV

Linearized V step (B.1) allows us to greatly reduce the cost of one V step. We ran Linearized PV on the quadratic objective (18) (Fig. 8).

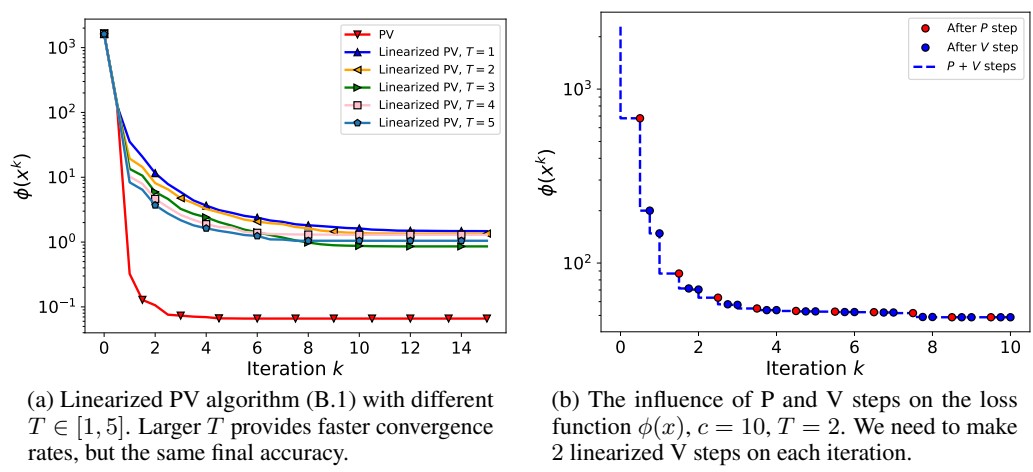

(a) Linearized PV algorithm (B.1) with different $T \in [1, 5]$. Larger $T$ provides faster convergence rates, but the same final accuracy.

(b) The influence of P and V steps on the loss function $\phi(x)$, $c = 10$, $T = 2$. We need to make 2 linearized V steps on each iteration.

Figure 8: Experiments with Linearized PV algorithm (B.1).

From the first experiment (Fig. 8a) we can see that

1. increasing $T$ provides slightly better convergence rates and approximately the same final accuracy. We saw the similar results on large-scale experiments. Hence, we can use $T = 1$ to save computations.

2. Linearized PV algorithm (B.1) converges to a worse accuracy than the exact PV (1).

3. Linearized PV has to make more iterations than PV to converge

The second plot (Fig.8b) demonstrates the effect of multiple Linearized V step. We can see that the largest effect comes from the first V step.

## Q.7  Linearized PV + sparse updates

In previous section (Q.6) we have seen that Linearized PV algorithm converges to a worse accuracy than the exact PV (1).

Linearized PV (B.1) with combination of sparse updates (3.3) is intended to mitigate this issue.

On (Fig. 9a) we can see comparison of three methods: exact PV (1) – red line, Linearized PV (B.1) – blue line and Linearized PV with sparse updates (3.3) – green line.

From this experiment we observe

1. Linearized PV with sparse updates converges to a better accuracy than Linearized PV

2. inearized PV + sparse updates has to make more iterations than Linearized PV and exact PV to converge

Hence, this approach helps us to converge to a better accuracy, but with a price of larger number of iterations to converge.

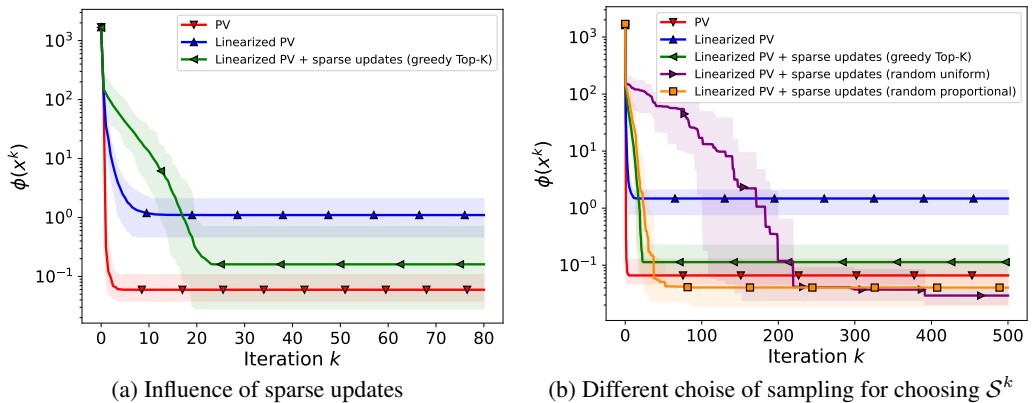

(a) Influence of sparse updates

(b) Different choise of sampling for choosing $\mathcal{S}^k$

Figure 9: Comparison of the exact PV algorithm (1) with Linearized PV (B.1) and Linearized PV + sparse updates (3.3).

We can use different rules for choosing the subspace $\mathcal{S}^k$ which produce different convergence rates and the final accuracy levels (9b).

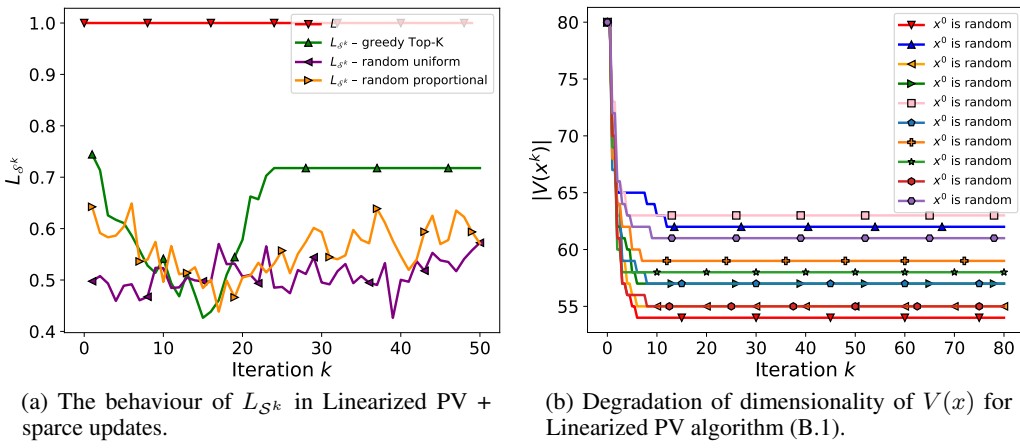

(a) The behaviour of $L_{\mathcal{S}^k}$ in Linearized PV + sparce updates.

(b) Degradation of dimensionality of $V(x)$ for Linearized PV algorithm (B.1).

Figure 10: Comparison of the exact PV algorithm (1) with Linearized PV (B.1) and Linearized PV + Sparse Updates (3.3).

In large scale experiments we used sparse updates with $\mathcal{S}^k$ being chosen greedily based on the absolute values of the entries of gradient vector $|\nabla_i \phi(y^k)|$ – green line. Here we demonstrated that other strategies based on random uniform sampling or random proportional to $|\nabla_i \phi(y^k)|$ sampling can be even more advanced providing better final accuracy at cost of larger number of iterations to converge – purple and orange lines (Fig. 9b).

We can see that Linearized PV with sparse updates can converge to even a better accuracy than the exact PV algorithm. The problem of Linearized PV algorithm is that we come to the local minimum and cannot get out of that minima because of the small value of the gradient (we are near to a real solution) and small stepsize.

Linearized PV with sparse updates allows us to mitigate this problem by reducing the subspace from $\mathbb{R}^d$ to $\mathcal{S}^k$ in which we are solving this optimization problem. This allows us to greatly reduce the local Lipschitz constant from $L$ to $L_{\mathcal{S}^k}$ and hence significantly increase the stepsize $\gamma = 1/L_{\mathcal{S}^k}$. We demonstrated how $L_{\mathcal{S}^k}$ changes for Linearized PV with different sparse updates sampling methods (Fig. 10a).

## R    PV$^+$ Algorithm

We can have degradation of $|V(x^k)|$ during both the P and V steps. Here are examples for both of them with optimizing simple quadratic objective $\phi(x) = \sum_{i=1}^{d} (x_i - x_i^\star)$ (case when all $a_i$, $i \in [d]$ in (18) are equal to one).

1. **Degradation during the P step:** Let $x^\star = \{0, 2, 1\}$, $x^0 = \{x, x, y\}$, so $|V(x^0)| = 2$, then after the P step we will have $y^0 = \{1, 1, 1\}$, hence $|V(y^0)| = 1$.

2. **Degradation during the V step:** Let $x^\star = \{2, 10, 0, 11\}$, $x^0 = \{x, y, y, z\}$, so $|V(x^0)| = 3$, then after the P step we will have $y^0 = \{2, 5, 5, 11\}$. Finally, after the V step we have $x^1 = \{2, 11, 2, 11\}$, so $|V(x^1)| = 2$.

In real experiments we observe decreasing of $|V(x)|$. You can observe this phenomena with Linearized PV algorithm (10b). As we can see we have a big decreasing of $|V(x)|$ during the first several iterations. We can use this gap to find even better solution with improved final accuracy and faster convergence rate.

To add additional unique element in $V(x)$ we considered the modification of V step: V$^+$ step, where we allow $|V(x)|$ to become larger up to some upper bound.

Let $\phi(\cdot)$ is a mapping $\phi(\cdot) : \mathbb{R}^d \to \mathbb{R}$ and the vector $x^\star \in \mathbb{R}^d$ be the optimal point: $\nabla \phi(x^\star) = 0$. Let the vector $x \in \mathbb{R}^d_{\leq c}$ and define the set $W(x, x^\star, c)$ such that it contains at most $c - |V(x)|$ unique elements from $V(x^\star)$ without elements from $V(x)$:

$$W(x, x^\star, c) = V(x^\star) \setminus V(x) : \quad |W(x, x^\star, c)| = c - |V(x)| \tag{22}$$

---

**Algorithm 8** PV$^+$ Algorithm

1: **Parameters:** starting point $x^0 \in \mathbb{R}^d_{\leq c}$, the optimal point $x^\star$, maximal number of distinct values $c$,
2: **for** $k = 0, 1, \ldots$ **do**
3: $\quad y^k = \arg\min_{y \in \mathbb{R}^d} \left\{ \phi(y) : P(y) \sqsupseteq P(x^{k-1}) \right\}$ $\hspace{3cm}$ (P step)
4: $\quad x^{k+1} = \arg\min_{x \in \mathbb{R}^d} \left\{ \phi(x) : V(x) \subseteq V(y^k) \bigcup W(y^k, x^\star, c) \right\}$ $\hspace{1cm}$ (Modified V step)
5: **end for**

---

**Theorem R.1.** *Assume $\phi$ is bounded below, and let $x^0 \in \mathbb{R}^d_{\leq \hat{c}}$. Then the algorithm PV$^+$ has the following guarantees*

(i) $y^k, x^k \in \mathbb{R}^d_{\leq \hat{c}}$ *for all $k \geq 0$,*

(ii) $\phi(x^{k+1}) \leq \phi(y^k) \leq \phi(x^k)$ *for all $k \geq 0$, and*

(iii) *the sequence $\{\phi(x^k)\}_{k \geq 0}$ converges to some value, which is smaller or equal to one produced by PV algorithm*

*Proof.* **Part (ii):** Since

$$x^{k+1} = \arg \min_{x \in \mathbb{R}^d} \left\{ \phi(x) : V(x) \subseteq V(y^k) \bigcup W(y^k, x^\star, c) \right\}$$

and because $x = y^k$ satisfies the constraint $V(x) \subseteq V(y^k) \bigcup W(y^k, x^\star, c)$, we conclude that $\phi(x^{k+1}) \le \phi(y^k)$.

The rest of the proof is identical to (A.1). □

## S   Broader Impact

The main impact of our work, both positive and negative, is in the ability to deploy higher-quality LLMs to run on memory-limited devices like desktops, laptops, and phones. On the positive side, this would allow practitioners to develop offline LLM applications (e.g. translate service), lower-latency chat assistants that are not dependant on network latency, or privacy-sensitive LLM applications where the user's private data never leaves their device. Furthermore, this can facilitate the creation of free open-source software based on LLMs by eliminating the need to maintain costly inference servers on the backend. Since phones are everywhere and LLMs are powerful general-purpose tools, PV-tuned models could significantly impact how the general population uses LLMs to complete tasks.

However, LLMs are still a dual-use technology with the potential for significant benefits and serious harm. Risks range from deliberate misuse (e.g. spam generation) and accidental misuse to negative economic side-effects. An upper bound on these risks is that PV tuning does not create new (potentially risky) LLM capabilities, merely making existing ones more accessible.

