# OpenReview forum: "PV-Tuning: Beyond Straight-Through Estimation for Extreme LLM Compression"
_NeurIPS.cc/2024/Conference — NeurIPS 2024 oral_

### Official Review · Reviewer_APHA · 2024-07-10

**Soundness:** 3
**Presentation:** 2
**Contribution:** 3
**Rating:** 5
**Confidence:** 4

**Summary:**

The paper provides a new QAT algorithm for extreme compression of LLMs under various discrete weight representations (including vector quantization,  uniform quantization). Traditional optimization techniques rely on straight-through estimation (STE) for optimization in the presence of discrete parameters. This paper provides an alternative to QAT called PV-tuning which resembles the EM algorithm, and performs alternating optimization.

When specialized to standard uniform quantization, the algorithm does the following: in the P step, the scale parameter is updated using backprop, and V step performs rounding to the nearest scaled integer. One important aspect of the algorithm is that the V step only  updates a few coordinates at a time. This is mainly done to ensure the algorithm doesn't get stuck at a sub-optimal point.

Experimental results show that this new optimization algorithm leads to better quality models than existing PTQ techniques such as QuIP, AQLM, GPTQ on diverse models. Experiments on Llama 2 7B also show that this technique improves upon STE. In terms of compute efficiency, the proposed algorithm is 1.5x slower than standard fine-tuning.

**Strengths:**

Improving STE is a very important problem and is key to making progress on extreme compression of LLMs.  The current paper aims to tackle this problem. So, I believe the broad direction is interesting to the community.

The proposed algorithm has an EM flavor, which is known to work well for estimating latent variables models. I found it quite interesting that the proposed algorithm seems to out-perform STE for 2-bit compression of Llama 7B. That being said, there are certain weaknesses in the empirical evaluation that needs to be addressed.

**Weaknesses:**

- **Clarity** The presentation (especially the algorithmic, technical details) in the paper needs to be significantly improved. Instead of the non-uniform quantization example used in the paper, I would suggest using a more relevant example such as uniform quantization or vector quantization. In its current form, I found it hard to wrap my head around how the proposed algorithm can be used for vector quantization and uniform quantization (I had to look into the appendix to figure this out). This is probably because there is no single framework that can unify all the weight representations. Different representations require different definitions of P, V.
  - without the clean presentation, it is very hard to really understand how the proposed technique compares with existing QAT algorithms such as STE, at a conceptual level.
- **Experimental Evaluation** Better experimental evaluation is needed to understand the real utility of the proposed approach. To be precise, the experiments in Table 2 compare PV tuning with PTQ techniques. But it is well known that QAT outperforms PTQ techniques. So, a better baseline to compare in Table 2 is QAT using STE (or other state of the art QAT techniques).
- **Related Work** Uniform quantization, as done by GPTQ and QuIP, has been well studied in the literature. And several QAT techniques that improve upon STE have been developed for uniform quantization (packages such as Pytorch, JAX, TF have really good implementations of QAT for uniform quantization). It's a bit surprising that none of these works are discussed in the paper.

**Questions:**

- It would be great if the authors can provide some explanation for why PV tuning outperforms STE for uniform quantization.
   - For the case of uniform quantization, it looks like PV tuning does some form of alternating minimization for learning the scale parameters and the quantized weights. In contrast, STE jointly optimizes for both scale parameters and quantized weights. Intuitively, shouldn't joint optimization perform better than alternating optimization?
- It looks like PV tuning only updates a few coordinates in each iteration. Doesn't this slow down the convergence of the algorithm and increase the training cost (for each backprop we would be updating very few coordinates)?
- If tau is set dynamically (as described in line 304), wouldn't it hurt the performance of optimizers like Adam which maintain coordinate-wise gradient statistics?
- It looks like in experiments, STE is applied on top of PV-tuning. But this algorithm is never described in the paper. It would be great if the authors can provide some details about this algorithm?

**Limitations:**

See weaknesses and questions.

---

> ### Author Rebuttal · Authors · 2024-08-07
>
> We thank the reviewer for their feedback and address their concerns below.
> > (Clarity) The presentation (especially the algorithmic, technical details) in the paper needs to be significantly improved.
>
> We agree with your comments on the presentation, and will try to make the algorithm definition easier to follow. Non-uniform scalar quantization was chosen for illustration as a non-trivial example with multiple continuous (1-d codebooks, locations of the quantization points) and discrete (1-d codes, codebook assignments) parameters, being at the same time simpler than multidimensional quantization. PV framework can be easily extended to various quantization formats. Specifically, for symmetric uniform scalar quantization there is only a single continuous parameter (P) - quantization scale and discrete parameters (V) are the integers on a line segment. For vector quantization, continuous parameters (P) form a set of vectors (codebooks), and discrete parameters (V) are the indices corresponding to a vector from the codebook.
>
> > (Experimental Evaluation) Better experimental evaluation is needed to understand the real utility of the proposed approach. To be precise, the experiments in Table 2 compare PV tuning with PTQ techniques. But it is well known that QAT outperforms PTQ techniques. So, a better baseline to compare in Table 2 is QAT using STE (or other state of the art QAT techniques).
>
> We provide a comparison with QAT methods based on STE and Stochastic Rounding in Table 1, Section 4.2. PV-tuning outperforms STE for all quantization methods considered (GPTQ, VQ, AQLM). Therefore, in the following comparison in Table 2 we adopted PV-tuning as the most performant fine-tuning strategy and focused on comparing against established baselines. However, if the reviewer sees a better way to arrange these results, we will welcome their suggestion (through an update on openreview).
> > (Related Work) Uniform quantization, as done by GPTQ and QuIP, has been well studied in the literature. And several QAT techniques that improve upon STE have been developed for uniform quantization (packages such as Pytorch, JAX, TF have really good implementations of QAT for uniform quantization). It's a bit surprising that none of these works are discussed in the paper.
>
> We agree with the concern and will update Section 2 to discuss these techniques, such as  [1, 2,3,4] and other works. To the best of our knowledge, PyTorch and TF use vanilla STE in their QAT implementation, which we have already compare against. We will investigate this further for the final version.
>
> Questions
>
> > (Q1) It would be great if the authors can provide some explanation for why PV tuning outperforms STE for uniform quantization. For the case of uniform quantization, it looks like PV tuning does some form of alternating minimization for learning the scale parameters and the quantized weights. In contrast, STE jointly optimizes for both scale parameters and quantized weights. Intuitively, shouldn't joint optimization perform better than alternating optimization?
>
> We do our best to explain this below and will incorporate this in the final version of the paper. When comparing linearized subspace PV (our practical algorithm) and STE, the main difference is that STE tries to update all weights on every step, whereas our algorithm only updates a chosen subset. We explain why updating all weights is problematic in L192-204. In L218-221 and Appendix G, we explain how choosing a subset of weights helps with discrete optimization. In practice, STE can improve the model during the first steps, but eventually diverges without reaching the optimal solution. Furthermore, as we describe in 787-788, PV-tuning can also do P and V steps simultaneously.
>
> > (Q2) It looks like PV tuning only updates a few coordinates in each iteration. Doesn't this slow down the convergence of the algorithm and increase the training cost (for each backprop we would be updating very few coordinates)?
>
> The V step in PV-tuning performs large updates on the discrete parameters. Constraining updates to a subset of parameters is necessary for stable convergence of the algorithm. Nevertheless, PV-tuning with part of the parameters updated exhibits steadier and faster convergence compared to STE
>
> > (Q3)If tau is set dynamically (as described in line 304), wouldn't it hurt the performance of optimizers like Adam which maintain coordinate-wise gradient statistics?
>
> We agree that updating a varying fraction of hyperparameters on each step makes the optimization problem more intricate. However, the preconditioning used in Adam and similar preconditioned gradient descent method is only a coarse approximation of a true curvature, thus the performance could be quite robust to inaccuracies in preconditioner estimates.
>
> > (Q4)It looks like in experiments, STE is applied on top of PV-tuning. But this algorithm is never described in the paper. It would be great if the authors can provide some details about this algorithm?
>
> PV-tuning is proposed as an alternative to STE for continuous and discrete optimization. In Table 1 we provide a comparison between PV-tuning and STE and show that PV-tuning outperforms STE.
>
>
> [1] Zhou, Shuchang, et al. (2016), arXiv:1606.06160
>
> [2] Chen, S. et al. (2019), Metaquant: Learning to quantize by learning to penetrate non-differentiable quantization. NeurIPS 2019
>
> [3] A. Shekhovtsov and V. Yanush (2020), arXiv:2006.06880
>
> [4] M. Spallanzani et al. (2022), arXiv:2203.11323

---

### Official Review · Reviewer_NP8P · 2024-07-11

**Soundness:** 4
**Presentation:** 4
**Contribution:** 3
**Rating:** 7
**Confidence:** 3

**Summary:**

This paper proposes a novel PV-tuning algorithm for extreme compression of LLMs. In each iteration, two steps (P step and V step) are performed, each aimed at reducing the loss function. Different than existing methods that use straight-through estimator (STE), PV-tuning employs a subspace search strategy at the V step to enable large updates and overcome the training stagnation caused by discreteness. Extensive experiments demonstrate that the proposed method achieve superior accuracy on models such as LLaMA and Mistral.

**Strengths:**

1. The proposed PV framework comes with theoretical guarantees.

2. The subspace descent in the V step is a novel approach to overcome the stagnation issue caused by discreteness, and it appears to work better than the popular STE method.

2. PV-tuning is validated through extensive experiments and consistently outperforms all existing methods for 1-bit and 2-bit vector quantization.

**Weaknesses:**

1. PV-tuning is computationally more expensive than existing PTQ methods and requires more GPU resources on larger models. Could you report the runtime of PV-tuning in your expiriments?

2. The authors claim that STE-based optimization is not well justified and leads to poor practical performance. However, I am aware of a paper [1] that provided the theoretical justification for STE approach.
  - [1] Yin el al. Understanding Straight-Through Estimator in Training Activation Quantized Neural Nets, ICLR 2019.

**Questions:**

1. In my view, the PV algorithm seems similar to alternating minimization. Is there a connection between these two algorithms?

2. Why is the P step an unconstrained minimization problem, as there is a constraint $P(y) \subseteq P(x)$?

3. Is the linearized V step described in section 3.2 equivalent to projected gradient descent? What is the computational cost to compute the projection onto {$x: V(x) \subseteq V(y)$} in the V step ?

---

> ### Author Rebuttal · Authors · 2024-08-07
>
> We thank the reviewer for their feedback and we are glad that they appreciate our technical contribution and experiments. Below we do our best to address their concerns and answer questions.
>
>
> > (W1) PV-tuning is computationally more expensive than existing PTQ methods and requires more GPU resources on larger models. Could you report the runtime of PV-tuning in your experiments?
>
> We acknowledge the fact that PV-tuning requires significant computation. PV-tuning is positioned between PTQ (post training quantization) methods (which do everything in one-shot on a small amount of data) and QAT (quantization-aware training) methods (which can require even more significant amounts of data and computation): we use little data, but need to be able to fine-tune full blocks or even the entire model. The improved accuracy should offset this additional cost.
>
> Yet, we stress that the runtime of the finetuning procedure is still negligible relative to LLM training. Specifically, PV-tuning of Llama-2-7b and Llama-2-70b on a single machine with 8 A100 GPUs took 90 hours and 171 hours, respectively (the 70b model was trained with a smaller number of iterations). This amounts to ~1000 gpu-hours.  While this is not negligible, our fine-tuning time is still orders of magnitude smaller than the time it took to train the model: for instance, training Llama-2-7b and Llama-2-70b model took 184’320 hours and 1’720’320 gpu-hours, respectively [3, 4].
>
> > (W2) The authors claim that STE-based optimization is not well justified and leads to poor practical performance. However, I am aware of a paper [1] that provided the theoretical justification for STE approach.
>
> The referenced work presents important insights about the application of STE to the training algorithm, together with the set of prescriptions for a proper choice of the STE. However, we believe that the setup in [1]  is very different from the one considered in our paper. Firstly, **this work applies STE for non-differentiable activations, rather than quantized model weights**. Moreover, their experiments are done on small scale networks and datasets (small MLPs and CNNs on MNIST/CIFAR10) and the conclusion may not transfer to LLMs and large datasets. There are, however, works that study the STE applied to weight quantization, such as [1, 2]: these works found that the standard (deterministic) STE does not converge to the optimum and propose alternatives that do converge, but introduce noise similar to the stochastic rounding that we evaluate in Section 4.2 (see Appendix F for details).
>
> > In my view, the PV algorithm seems similar to alternating minimization. Is there a connection between these two algorithms?
>
> Yes, the PV-Tuning algorithm can be seen as a particular case of alternating minimization / coordinate descent, where one alternates between optimization of discrete and continuous parameters.
>
> > Why is the P step an unconstrained minimization problem, as there is a constraint $P(y) \subseteq P(x)$?
>
> We meant that the optimization problem for P-step can be reparameterized as an unconstrained optimization problem on the $c$ unique values in the weight matrix. From a practical viewpoint, we declare the $c$ “clusters” or codebooks as an optimized variable, and use an automated differentiation engine (i.e. PyTorch) to minimize the loss as a function of these values. We will clarify this in (L161) in the next revision of the paper.
>
> > Is the linearized V step described in section 3.2 equivalent to projected gradient descent? What is the computational cost to compute the projection onto {$x: V(y) \subseteq V(x)$} in the V step?
>
> There is indeed some similarity to the projected gradient descent, applied to a discrete space of quantization codes. In this interpretation, the projection operation is finding the nearest quantized value (in terms of L2) out of a fixed-size set. For non-vector quantization (e.g. GPTQ in Section 4.2), this projection is done by simple rounding, which is a constant time per weight, after accounting for the quantization scale and zero point (see L294). As for the vector quantization, the projection is done by searching the set of all vectors in the current quantization “codebook” (L295-296). Note that this projection is only computed for a small fraction of weights in the current subspace (Section 3.3).
>
> [1] A. Shekhovtsov and V. Yanush. Reintroducing straight-through estimators as principled methods for stochastic binary networks. In DAGM German Conference on Pattern Recognition, pages 111–126. Springer, 2021.
>
> [2] M. Spallanzani, G. P. Leonardi, and L. Benini. Training quantised neural networks with STE
> variants: the additive noise annealing algorithm. In Proceedings of the IEEE/CVF Conference
> on Computer Vision and Pattern Recognition, pages 470–479, 2022.
>
> [3] Touvron, Hugo, et al. "Llama 2: Open foundation and fine-tuned chat models." arXiv preprint arXiv:2307.09288 (2023).
>
> [4] Dubey, Abhimanyu, et al. "The Llama 3 Herd of Models." arXiv preprint arXiv:2407.21783 (2024).

---

> > ### Comment · Reviewer_NP8P · 2024-08-12
> >
> > Thank you for answering my questions. After reading the rebuttal, I keep my initial rating.

---

### Official Review · Reviewer_p3Lv · 2024-07-12

**Soundness:** 2
**Presentation:** 1
**Contribution:** 3
**Rating:** 5
**Confidence:** 2

**Summary:**

This paper proposes a new PTQ fine-tuning algorithm called PV tuning. Recent SOTA LLM PTQ works include fine tuning steps to recover the original model on top of the actual quantization step. Fine tuning has been shown to be an effective and relatively cheap to run (vs full QAT) way to improve quantization quality. However, fine tuning for LLM PTQ has not been well explored, in part due to it only being recently introduced and also LLMs being expensive to do things with. My understanding of PV tuning is that it performs alternating optimization on a quantized LLM. Specifically, it alternates between optimizing continuous parameters (codebook values, layernorms, etc) and what is essentially a form of coordinate descent on codebook assignments to weight matrix entries.

**Strengths:**

- PV tuning appears to achieve strong empirical results on a wide variety of models. Notably, PV tuning achieves good performance on Llama 3, which GPTQ-based methods tend to do poorly on.
- PV tuning does not seem too costly to run vs existing fine tuning methods and should be compatible with a wide range of "base" quantization methods.

**Weaknesses:**

- This paper is written in a dense and arcane way and is hard to follow. While there are some theorems, they do not appear to be very useful (eg 3.1 just claims that PV tuning converges, but without any statements on how good the final solution is or any bounds on error). Perhaps I am misunderstanding the paper completely, so it would be useful for the authors to give a TLDR version of the actual method.
- How large is the subspace for the CD step? Iterating through all possible code assignment changes is exponential in the subspace size, so how do you perform the code assignment optimization in a tractable way?
- The empirical results presented are based off of the AQLM quantization method. One main benefit of quantizing LLMs is to achieve speedups in memory bound scenarios. However, the AQLM configuration used for PV tuning is too slow for fast inference (relative to what the hardware *could* achieve) due to the size of the codebook. How well does PV tuning perform with a smaller codebook (eg a 10 bit 8D codebook, which would use 16KB)? The paper also mentioned PV tuning could be applied to QuIP#, which does achieve near memory bandwidth. Do you have any experiments showing this? QuIP# is based off of GPTQ which performs poorly on Llama 3, does PV tuning fix this if applied to QuIP# after GPTQ?
- I started reviewing this paper a few weeks ago and lost the first copy I was marking up, but I remember there were some inconsistencies in the result tables with numbers reported in the baselines, and some of the baseline results in 4.1 seemed a bit suspicious. Please check that your numbers are indeed correct. The pareto optimality claim is also questionable, since the "frontier" is just for PV tuning. A better comparison would be PV tuning @ 2 bits vs FP16 @ 3 bits, and it doesn't seem (although I didn't plot this) that PV tuning is actually better at 2 bits than FP16 at 3 bits.

**Questions:**

see above

**Limitations:**

yes

---

> ### Author Rebuttal · Authors · 2024-08-07
>
> We thank the reviewer for their feedback and address their concerns below.
>
> > (W1) This paper is written in a dense and arcane way and is hard to follow. While there are some theorems, they do not appear to be very useful (e.g. 3.1 just claims that PV tuning converges, but without any statements on how good the final solution is or any bounds on error). Perhaps I am misunderstanding the paper completely, so it would be useful for the authors to give a TLDR version of the actual method.
>
> While our main results are practical, we agree that our theoretical results can be presented more clearly, and we will do our best to improve presentation. For instance, Theorem 3.1 proves that PV-Tuning converges to a stable solution in order to contrast our method against Straight Through Estimation and Stochastic Rounding, both of which lack this property. We agree that proving tight convergence bounds would be valuable, but it is difficult to prove such bounds for our mixed continuous-discrete optimization problem–intuitively, the additional constraints make the problems “harder” than standard convex/non-convex optimization. We see this as an important direction for future work.
>
> > (W2) How large is the subspace for the CD step? Iterating through all possible code assignment changes is exponential in the subspace size, so how do you perform the code assignment optimization in a tractable way?
>
> The size of the subspace $\tau$ is such that the update satisfies  $\|x^{k+1} - x^k\| / \|x^k\| \leq 0.01$, also known as the trust ratio (see Section 3.3). We choose this subspace using a simple heuristic: we select $\tau$ parameters with the largest gradient of the original loss function. Further details can be found in Section 3.4. In practice, choosing the subspace takes negligible time, compared to accumulating the gradients w.r.t. LLM parameters.
>
>
> > (W3) The empirical results presented are based on the AQLM quantization method. One main benefit of quantizing LLMs is to achieve speedups in memory bound scenarios. However, the AQLM configuration used for PV tuning is too slow for fast inference
>
> We have also presented results for, e.g. the GPTQ format in our paper, although we indeed focused primarily on AQLM as it is one of the best methods in terms of model size vs accuracy. Fortunately, VQ and AQLM still offer speedups in memory-bound scenarios. We discuss this briefly in L334-337 and report speedups in Appendix M.
>
> > (W3) Do you have any experiments showing this? QuIP# is based off of GPTQ which performs poorly on Llama 3, does PV tuning fix this if applied to QuIP# after GPTQ?
>
>
> We have chosen AQLM over QuIP# because it outperformed the latter in WikiText2 perplexity (fig. 2). We have looked into integrating with QuIP#’s code base, but, due to its complexity, we are afraid that it won’t be possible to evaluate QuIP#’s performance with PV-tuning due to the rebuttal’s time constraints. We will try to add results in this direction for the next revision.
>
> > (W4) The result tables with numbers reported in the baselines, and some of the baseline results in 4.1 seemed a bit suspicious. Please check that your numbers are indeed correct.
>
> We would appreciate it if the reviewer could make this concern more specific. Below we respond based on our best guess about what the review refers to.
>
> There are two important causes why baseline numbers may seem odd. First, some of the baseline papers, notably [1] use a different way of computing the perplexity score, which is not compatible with most prior works. To compare these approaches, we reevaluated the methods from that paper using their official code. Second, for every paper that uses a different dataset or less data, we reevaluated these methods with the same sample of RedPajama we used (excluding OneBit, see L874-879). Please see Appendices J and L for a detailed experimental configuration. We verify all numbers that we report as a standard procedure, and will verify them again in the final revision.
>
>
>
> > (W4) The pareto optimality claim is also questionable, since the "frontier" is just for PV. A better comparison would be PV tuning @ 2 bits vs FP16 @ 3 bits, and it doesn't seem (although I didn't plot this) that PV tuning is actually better at 2 bits than FP16 at 3 bits.
>
> We use the same method for evaluating pareto optimality as in [2], the same method is used in follow-up works. Unfortunately, we don’t understand what “FP16 @ 3 bits” stands for. We encourage the reviewer to clarify their suggestion (e.g. by editing their review) so we can apply it in the final version of the paper.
>
>
> [1] Huang, W., et al. arXiv preprint arXiv:2404.14047 (2024).
>
> [2] Dettmers, T., Zettlemoyer L. arXiv preprint arXiv:2212.09720 (2022).

---

> > ### Comment · Reviewer_p3Lv · 2024-08-10
> > **more comments**
> >
> > - Regarding subspace selection, how do you do the code updates in a tractable way after selecting the subspace? Perhaps I am missing something, but this is still a discrete optimization problem after subspace selection, just a much smaller one. Is the "naive solution" that searches over all combinations cheap enough to do here?
> >
> > - Your main rebuttal said there were more experiments here with smaller codebooks, but I'm not seeing any. Did you forget to post part of the response? I'm interested in seeing how PV-tuning would work with a structured codebook, such as one that has sign symmetry. The 8 bit codebook result in the paper performs around the same as QuIP#, so it does appear that the quantizer quality still has an impact after PV-tuning.
> >
> > - Regarding the baselines, IIRC there were some FP16 numbers that didn't match. I don't have time to go back and check, so I will take your word that the tables are correct.
> >
> > - FP16 @ 3 bits just means if you plotted the FP16 perplexity numbers assuming the models were 3/16th the size. This represents what a lossless 3 bit model would achieve. Otherwise, if the 3 bit version of your quantizer just happens to be bad, then it would be very easy to outscale that.

---

> > > ### Author Response · Authors · 2024-08-11
> > > **Response to "more comments"**
> > >
> > > We are glad that the reviewer shows interest in our work and do our best to answer further questions:
> > >
> > > > Regarding subspace selection, how do you do the code updates in a tractable way after selecting the subspace? Perhaps I am missing something, but this is still a discrete optimization problem after subspace selection, just a much smaller one. Is the "naive solution" that searches over all combinations cheap enough to do here?
> > >
> > > In general (for VQ), we do an exhaustive search for each code in the chosen subspace, but **not all combinations**. This is because our Linearized V step (Section 3.2) minimizes mean squared error with the adjusted weight matrix (Equation 5). This problem can be solved independently for each discrete code because the optimization objective is a sum over independently chosen discrete weights.
> > >
> > > Still, this can be a costly procedure for LLMs with billions of parameters. However, it *becomes cheap since we only update a small fraction of codes*, i.e. the subspace. Thus, we run this costly optimization (1) for less than 1% of all weights and (2) in parallel on GPUs, making it practically feasible.
> > >
> > > In the supplementary code, the discrete optimization takes less than 10% of the total training time when running on A100s. However, further improvement of this algorithm is an interesting direction for future research.
> > >
> > > > Your main rebuttal said there were more experiments here with smaller codebooks, but I'm not seeing any. Did you forget to post part of the response? I'm interested in seeing how PV-tuning would work with a structured codebook, such as one that has sign symmetry. The 8-bit codebook result in the paper performs around the same as QuIP#, so it does appear that the quantizer quality still has an impact after PV-tuning.
> > >
> > > We thank the reviewer for finding this out, as this is an important suggestion. We accidentally did not include these numbers when compiling the response. You may find our response below:
> > >
> > > We conducted additional experiments using this configuration for Llama 2 7B in the same setup as the rest of our experiments (see Section 4.2). Specifically, we use a single 10-bit codebook with a group size of 8 consecutive weights, which leads to **1.26 bits per weight**, including codebooks. The resulting quantized model has a perplexity of 6.82 on WikiText-2 and 8.82 on C4. We compare these results with the previous configuration below:
> > >
> > > | Method | Avg Bits | Wiki2 $\text{PPL}\downarrow$ | C4 $\text{PPL}\downarrow$ | ArcC $\text{Acc.}\uparrow$ | ArcE $\text{Acc.}\uparrow$ | HellaSwag $\text{Acc.}\uparrow$ | PiQA $\text{Acc.}\uparrow$ | WinoGrande $\text{Acc.}\uparrow$ | Average $\text{Acc.}\uparrow$ |
> > > |---|---|---|---|---|---|---|---|---|---|
> > > | $-$ | 16 | 5.12 | 6.63	| 43.43 | 76.3 | 57.14 	| 78.07 | 69.06 | 64.80 |
> > > | PV-Tuning | 1.58 | 7.32 | 9.35 | 25.85 | 57.58 | 40.88 | 68.99 | 57.77 | 50.21 |
> > > | PV-Tuning (10-bit codebook) | 1.26 | 6.82 | 8.82 | 32.94 | 68.27 | 49.01 | 74.76 | 64.17 | 57.83 |
> > >
> > > **These are very promising results.**, we originally opted for 8- and 16-bit codes as they are easier to handle in modern hardware, but in principle, it is possible to pack 10-bit codes in a way that supports efficient inference. Following your suggestion, we will evaluate more configurations with small codebooks in the final version. Likewise, we will further explore structured codebooks as the reviewer suggested.
> > >
> > > > FP16 @ 3 bits just means if you plotted the FP16 perplexity numbers assuming the models were 3/16th the size. This represents what a lossless 3 bit model would achieve. Otherwise, if the 3 bit version of your quantizer just happens to be bad, then it would be very easy to outscale that.
> > >
> > > Thank you for clarifying this. By this upper bound criterion, the Pareto-optimality would indeed be achieved at a somewhat higher bitwidth. In our submission, we used a popular criterion from prior works, but your upper bound criterion could bring more nuance to the Pareto-optimality. We will add the comparison to FP16 @ 3 bits in the final version of the paper.

---

> > > > ### Comment · Reviewer_p3Lv · 2024-08-11
> > > > **Thanks**
> > > >
> > > > Thanks for the additional information. I am happy with the author's responses and will keep my score.

---

### Official Review · Reviewer_5n4W · 2024-07-12

**Soundness:** 3
**Presentation:** 3
**Contribution:** 3
**Rating:** 6
**Confidence:** 4

**Summary:**

The paper focuses on fine-tuning techniques over compressed weights and proposes PV-Tuning, a general framework that improves existing fine-tuning strategies and provides convergence guarantees. Experiments show that PV-Tuning outperforms prior techniques and achieves the first Pareto-optimal quantization for Llama-2 models at extreme compression. The idea is novel and reveals the importance of fine-tuning strategies.

**Strengths:**

The technical contribution of the work seems solid. In particular, the authors conduct extensive experiments to demonstrate the effectiveness of PV-Tuning coupled with the state-of-the-art quantized representations.

**Weaknesses:**

1. The paper introduces the principle and specific implementation of PV-Tuning in detail, but does not compare the differences in principle and specific implementation with the existing STE method and Stochastic Rounding method. It would be beneficial to include such a comparison for clarity.
2. Some notations in the paper may be confusing. In part 3.1, the expression “P(y) ⊇ P(x)” is used but it does not mean that P(x) is a subset of P(y), which is confusing because the ⊇ symbol usually means to include. Also in this part, it would be clear if there are some simple examples.

**Questions:**

1. In the experiment part, besides the PPL and accuracy, is there any evidence that supports the STE is noisier than PV-Tuning? It would be more convincing if the paper could provide some experimental evidence.
2. From Table 1, it seems the naïve Linearized PV without subspace doesn’t achieve pleasing performance while subspace Linearized PV performs well. Does this mean that subspace is the key to performance improvement? Can you combine subspace and STE to conduct some ablation experiments?

**Limitations:**

Please see the Weaknesses and Questions.

---

> ### Author Rebuttal · Authors · 2024-08-07
>
> We are glad that the reviewer appreciates our technical contribution and extensive experiments. We do our best to address their concerns and answer questions below.
>
> > (W1) The paper introduces the principle and specific implementation of PV-Tuning in detail, but does not compare the differences in principle and specific implementation with the existing STE method and Stochastic Rounding method. It would be beneficial to include such a comparison for clarity.
>
> Currently, we describe each of these approaches in appendices E and F respectively (L676–717), but we do not compare them directly in terms of the underlying principles and specific implementations.
>
> To address this concern, we will add a dedicated comparison between these two strategies and PV-tuning. More specifically, we will highlight the additional memory requirements of STE and PV-tuning, compared to stochastic rounding, and the effects of the stochastic rounding noise on the optimization. As for the experimental differences, we compared PV-Tuning vs the two other baseline strategies in Section 4.2. We would welcome further suggestions for comparisons.
>
> > (W2) Some notations in the paper may be confusing. In part 3.1, the expression “$P(y) \supseteq P(x)$” is used but it does not mean that P(x) is a subset of P(y), which is confusing because the $\supseteq$ symbol usually means to include. Also in this part, it would be clear if there are some simple examples.
>
> We do our best to formally define our notation, e.g. we explain the meaning of “$P(y) \supseteq P(x)$” in L127-128. However, we agree that it would be better to avoid collision with existing set notation and will do so in the next revision. We currently consider changing this to $\sqsupseteq$, but we would welcome additional suggestions. In the next revision of the paper we will also add examples that demonstrate this relation.
>
> > (Q1) In the experiment part, besides the PPL and accuracy, is there any evidence that supports the STE is noisier than PV-Tuning? It would be more convincing if the paper could provide some experimental evidence.
>
> In preliminary experiments we observed that training STE results in model diverging after a certain number of steps, with point of divergence varying with random seed. To better quantify this effect, we have run several experiments with different random seeds and reported std and error bars in **Figure 2** in the attached PDF.
> >(Q2) From Table 1, it seems the naïve Linearized PV without subspace doesn’t achieve pleasing performance while subspace Linearized PV performs well. Does this mean that subspace is the key to performance improvement? Can you combine subspace and STE to conduct some ablation experiments?
>
> Indeed, the subspace optimization during V step is the key to our performance improvements. From a practitioner’s standpoint, we introduce the general PV framework in order to show why and how subspace descent during V step can help with the optimization. We elaborate on this reasoning in lines 208-210 and 224-227.

---

> > ### Comment · Reviewer_5n4W · 2024-08-08
> >
> > Thank the authors. After reading the rebuttal, I keep my initial scoring.

---

### Official Review · Reviewer_3n7c · 2024-07-13

**Soundness:** 3
**Presentation:** 2
**Contribution:** 3
**Rating:** 5
**Confidence:** 3

**Summary:**

The paper proposes the use of fine-tuning over highly-compressed models to achieve better model compression. By introducing the PV-tuning algorithm, the authors have achieved superior discrete parameter optimization compared to traditional STE methods. The effectiveness of PV-tuning has been validated by the authors using various base models and datasets.

**Strengths:**

1. The authors have tested the effectiveness of PV-tuning under different experimental settings. The improvements are notably significant compared to the baselines.
2. The authors have found that fine-tuning should be employed to achieve better model compression rather than one-shot quantization.
3. The authors have proposed the PV-tuning method, addressing the issue of small gradient updates in discrete optimization.

**Weaknesses:**

1. I would appreciate it if the authors could clarify the distinction between fine-tuning and one-shot quantization as mentioned in lines 37-43. What exactly does "one-shot" imply? Does it refer to layer-wise quantization? Conversely, what does the authors' claim of "fine-tuning layer-wise" mean? As I understand, quantization typically uses a small calibration set, yet the authors have utilized the entire RedPajama for calibration in section 4.1. Is the "fine-tuning" mentioned by the authors simply calibration on a larger scale?

2. It seems the main problem the authors aim to address (line 45) is the further fine-tuning of already highly-compressed models on a large-scale dataset (RedPajama), rather than the traditional quantization/compression methods applied to full precision models. I agree that this approach might be effective. However, if this is the case, there might be an issue with the experimental setup in 4.1. Firstly, Figure 2 should compare the effects of each method with and without fine-tuning; the current Figure 2 middle and right do not directly facilitate this comparison. Secondly, the authors should not only select quantized representations from different methods for fine-tuning but also apply different fine-tuning strategies, including STE, PV-tuning, GPTQ, etc., to demonstrate the advantages of PV-tuning.

3. In Table 2, PV-tuning indeed shows a significant effect. However, I am uncertain whether this is due to PV-tuning using more data (RedPajama) or because it is inherently more effective. Did the authors use the same calibration data for different methods in this experiment?

**Questions:**

An important question: if we need to perform large-scale fine-tuning to achieve model quantization, why not maintain full precision during this large-scale fine-tuning and only apply low-precision quantization techniques after full precision fine-tuning?

**Limitations:**

Yes

---

> ### Author Rebuttal · Authors · 2024-08-07
>
> We are glad that the reviewer appreciates our technical contribution and accuracy improvements. Below, we do our best to address concerns and answer questions raised in the review.
>
> > (W1) What does “one shot” imply? Does it refer to layer-wise quantization?
>
> Indeed, in L37-38, one-shot quantization refers to the family of quantization algorithms that compress the model in a single layer-by-layer pass over the model, without fine-tuning. Typically, this is done through the use of advanced quantized representations. This includes layerwise quantization algorithms such as GPTQ [1], SPQR [2], or QUIP [3], among others, and is contrasted to algorithms that fine-tune the entire model or a subset of the layers, such as OmniQuant [4], QuIP#[5] or AQLM[6].
>
> > (W1) Conversely, what does the authors' claim of "fine-tuning layer-wise" mean?
>
> This refers to a type of algorithm that fine-tunes every transformer block individually, relative to the outputs of the original model, to minimize MSE between original and quantized outputs. This technique is implemented in, e.g., AQLM [6] and QUIP# [5].
>
> > (W1) As I understand, quantization typically uses a small calibration set, yet the authors have utilized the entire RedPajama for calibration in section 4.1.
>
> We believe there is a misunderstanding: as we describe in the dataset configuration (Appendix H, referenced in L243), **we use not the entire RedPajama dataset, but a small sample (0.1%) for that dataset**. Thus, we use a similar amount of calibration data to prior popular quantization techniques, such as AQLM and QuIP#.
>
> We agree that some of these terms (e.g. “one-shot”) and details can be described more accurately. We will rewrite L37-38 and L243 to make our setup easier to follow.
>
> > (W2) Figure 2 should compare the effects of each method with and without fine-tuning; the current Figure 2 middle and right do not directly facilitate this comparison.
>
> Thank you for the suggestion. We conducted additional experiments for some of the missing methods (e.g. GPTQ) and reported all results in **Table 1 and Figure 1** in the attached PDF. Note that one of the algorithms (SpQR) is still missing since the algorithm’s codebase requires additional work to incorporate proper finetuning. We will rectify this by the final version of the paper.
>
> In the next revision, we will improve Figure 2 to make it easier to compare each of these methods with and without finetuning.
>
> > (W2) The authors should not only select quantized representations from different methods for fine-tuning but also apply different fine-tuning strategies, including STE, PV-tuning, GPTQ, etc., to demonstrate the advantages of PV-tuning.
>
> We compare different fine-tuning strategies, including STE, PV-tuning, stochastic rounding and some of their combinations, in Section 4.2, Table 1. We apply each of these strategies to both GPTQ, VQ and AQLM to better understand their behavior for different representations. If you have any further suggestions to improve this analysis, we would be happy to apply them.
>
> > (W3) In Table 2, PV-tuning indeed shows a significant effect. However, I am uncertain whether this is due to PV-tuning using more data (RedPajama) or because it is inherently more effective. Did the authors use the same calibration data for different methods in this experiment?
>
> PV-Tuning only trains on a small sample of the RedPajama data, similarly to the baselines.
> We use an official sample of the RedPajama dataset and compare algorithms in equal conditions, with a few exceptions. Specifically, all algorithms except OneBit use samples  from the RedPajama dataset. For some of these algorithms (QuIP# and one-shot algorithms), their implementation cannot handle as much data within 1TB RAM due to storing activations in memory. For these algorithms, we used the largest possible dataset size and tested that the algorithm reaches saturation (i.e. adding data does not improve accuracy). Finally, the OneBit baseline uses a custom dataset described in its original paper. This is because the code for that specific algorithm was not available at the time and we had to use the existing results. We describe these details further in appendices H and L.
>
> > (Q1) If we need to perform large-scale fine-tuning to achieve model quantization, why not maintain full precision during this large-scale fine-tuning and only apply low-precision quantization techniques after full precision fine-tuning?
>
> Quantized fine-tuning works because it allows layers to adjust each other’s quantization errors. In contrast, fine-tuning an uncompressed model in full precision (e.g. by cross-entropy) may improve general model quality, but it does not compensate for the quantization error. Note that finetuning is much shorter than pretraining of the original model.
>
> To better demonstrate the difference between the two strategies, we run an additional experiment where we fine-tune the model in full precision and quantize it afterward. For this experiment, we substitute the KL divergence loss with cross-entropy on the data. This is because the KL divergence between two identical full precision models is always zero. We run this experiment with the Llama 2 7B model using 2-bit vector quantization in the same setup as in Section 4.3, and then quantize the model in one shot using AQLM. The results in Table 2 (PDF) show that this achieves some improvement over AQLM, but PV-Tuning still significantly outperforms this baseline. We hypothesize that the advantage of PV-Tuning is explained by the quantization error compensation during fine-tuning.
>
> [1] Frantar, E., et al. arXiv preprint arXiv:2210.17323 (2023).
>
> [2] Dettmers, T., et al. arXiv preprint arXiv:2306.03078 (2023).
>
> [3] Chee, J., et al. arXiv preprint arXiv:2307.13304 (2024).
>
> [4] Shao, W., et al. arXiv preprint arXiv:2308.13137 (2024).
>
> [5] Tseng, A., et al. arXiv preprint arXiv:2402.04396 (2024).
>
> [6] Egiazarian, V., et al. arXiv preprint arXiv:2401.06118 (2024).

---

> > ### Comment · Reviewer_3n7c · 2024-08-11
> >
> > Thanks for your responses. Another questions: have you compared the efficiency of PV-Turning with STE or other baselines?

---

> > > ### Author Response · Authors · 2024-08-11
> > > **Response to official comment by reviewer 3n7c**
> > >
> > > In terms of computation efficiency, PV-Tuning is roughly on par with STE since both of them are bottlenecked by accumulating gradients w.r.t. dequantized LLM weights. PV-Tuning with subspaces is slightly more efficient because it only computes V step for a small portion of parameters, but otherwise, the total runtime is comparable. The memory efficiency of STE and PV-tuning are equivalent. We discuss efficiency further in Appendix I for calibration and Appendix M for inference. Alternatively, if you meant efficiency as model accuracy, we analyze this in Section 4.2.

---

### Author Rebuttal · Authors · 2024-08-07

We thank the reviewers for providing valuable comments and suggestions. We are glad that the reviewers appreciate our strong empirical results (3n7c, 5n4w, p3Lv, NP8P) including the latest models (p3Lv). Reviewers also highlight our technical contribution (5n4w, 3n7c, NP8P) and theoretical guarantees (NP8P).

We provide an overview of the individual responses below:

**(R3n7c) Comparing the effects of each method with and without fine-tuning**

To facilitate this comparison, we evaluated the algorithms missing from Figure 2 of our submission (GPTQ and SpQR) and rearranged their presentation. We present these results in two ways. First, we report the performance of each method with and without fine-tuning in Table 1 (PDF). We also provide plots where we compare all methods, with and without fine-tuning, to provide global context (Figure 1 in response PDF). The reviewer also suggests additional experiments with different fine-tuning methods, which we address in the individual response.

**(R3n7c) why not maintain full precision during this large-scale fine-tuning and only apply low-precision  quantization techniques after full precision fine-tuning?**

While we detail our reasoning in the direct response, we generally agree that it is interesting to test this empirically. To that end, we conducted full-precision fine-tuning and quantized the fine-tuned model as requested. The resulting perplexity scores can be found in Table 2 in the attached PDF (3n7c). Overall, it appears that SoTA LLMs are already well tuned on general data (that we use for calibration); fine-tuning them further lead to negligible improvements on some tasks to the small detriment of others. In turn, PV-Tuning also does not improve the original model quality, but it allows the quantized layers to “re-adjust” to each other’s quantization errors, leading to better accuracy. We thank the reviewer for this interesting suggestion.

**(R5n4W) Verifying the observation that STE is noisier than PV-Tuning**

To better quantify this effect, we ran each algorithm 5 times with different random seeds  and reported the adjusted standard deviation for each iteration. Since running with multiple random seeds takes a lot of time, we chose to run these experiments using a smaller TinyLlama-1.1B LLM using 2-bit compression.

For convenience, we arrange our results in a plot **(Figure 2 in the response PDF)**, where we report **WikiText-2 perplexity every 5 training steps**. The solid line represents the mean perplexity, and the surrounding pale “error bars” represent standard deviation (with the same scale). In this experiment, we use the same data and experimental configuration as in Section 4.2 and provide some additional description in the figure caption (see PDF). To summarize, STE shows similar early performance, but eventually diverges with an increasing noise magnitude.

**(p3Lv) How well does PV tuning perform with a smaller codebook (eg a 10 bit 8D codebook, which would use 16KB)?**

We reported a similar small 8-bit codebooks in the original submission (Section 4.2, Table 1, right). To further address the reviewer’s concern, we evaluate the requested configuration (a single 10 bit codebook with group size 8, about 1.25 bits per weight) with Llama 2 7B, using the same data and hyperparameters as in Section 4.2. We report the resulting metrics in the individual response to p3Lv and welcome further suggestions.

We hope that these new results alleviate reviewers’ concerns. If not, we encourage reviewers to comment and clarify their requests in the author-reviewer discussion phase.

---

> ### Comment · Reviewer_p3Lv · 2024-08-10
> **full precision fine-tuning?**
>
> > (R3n7c) why not maintain full precision during this large-scale fine-tuning and only apply low-precision quantization techniques after full precision fine-tuning?
>
> > While we detail our reasoning in the direct response, we generally agree that it is interesting to test this empirically. To that end, we conducted full-precision fine-tuning and quantized the fine-tuned model as requested. The resulting perplexity scores can be found in Table 2 in the attached PDF (3n7c). Overall, it appears that SoTA LLMs are already well tuned on general data (that we use for calibration); fine-tuning them further lead to negligible improvements on some tasks to the small detriment of others. In turn, PV-Tuning also does not improve the original model quality, but it allows the quantized layers to “re-adjust” to each other’s quantization errors, leading to better accuracy. We thank the reviewer for this interesting suggestion.
>
> I'm having trouble understanding why this is a sensible thing to do / how such a setup would work. My understanding was that the goal of PV-tuning was to recover the original model through fine-tuning by tuning the quantized model on a calibration set. Before quantization, the model will always match itself on any dataset. Does this mean the tuning in PV-tuning actually minimizes autoregressive error on a dataset (eg perplexity), or does it minimize the error to the original model on a dataset? If its the former, then isn't PV-tuning essentially doing QAT to a *new* task?

---

> ### Author Response · Authors · 2024-08-11
> **Response to "full precision fine-tuning?"**
>
> Your original understanding is correct: PV-tuning aims to minimize the difference between original and compressed model output in terms of KL divergence, not the autoregressive loss. Another reviewer asked us to run an additional ablation experiment in a different setup. The meaning of this experiment requires context from our discussion with (R3n7c):
>
> > An important question: if we need to perform large-scale fine-tuning to achieve model quantization, why not maintain full precision during this large-scale fine-tuning and only apply low-precision quantization techniques after full precision fine-tuning?
>
> We conduct this experiment as a sanity check, to answer the question: how would the model fare *if we were to maintain full precision* during fine-tuning. In this (different) setup, it makes no sense to minimize KL (as you said), and for this reason, **we only minimize the autoregressive cross-entropy in this side-track experiment**.
>
> To the best of our understanding, the reviewer may have suggested the experiment as an extra ablation.
>
> We hope this clarifies the reviewer’s question, but we are happy to discuss this further.

---

> > ### Comment · Reviewer_p3Lv · 2024-08-11
> > **Makes sense**
> >
> > Thanks for clarifying. I guess all that experiment shows is that the fine-tuning dataset isn't the same as the pretraining dataset, which is obvious.

---

### Decision · Program_Chairs · 2024-09-25

**Decision:**

Accept (oral)

**Comment:**

The paper considers compression of LLMs through quantization, focusing on fine-tuning techniques over initially compressed weights so as to better approximate the performance of the unquantized model. The method PV-Tuning is developed and compared with existing baselines.

The reviewers have noted that the method is novel, reveals the importance of discovering good fine-tuning strategies, extensive experiments validate the method and show improvement on baseline methods, and achieves good performance on Llama 3 (where GPTQ-based methods do not).

The optimization is a mixed continuous-discrete case that complicates the solution. The PV-Tuning method is compared with and shown to outperform straight through estimation (STE), and some new insights into the problem and methods is gained as well.

Although significant, the PV-tuning method seems to have complexity relative to existing fine-tuning methods and should be compatible with various initial quantization methods.

The paper provides some convergence analysis and guarantees, contrasting with straight through estimation (STE) and stochastic rounding that generally lack such guarantees.  The reviewers noted that the performance analysis was not always clearly presented.

The rebuttal and discussion with reviewers has led to further interesting experiments and potential algorithmic improvements.